# Widespread reorganisation of pluripotent factor binding and gene regulatory interactions between human pluripotent states

Peter Chovanec[1,2,6], Amanda J. Collier [3,6], Christel Krueger [3], Csilla Várnai [2,4], Claudia I. Semprich [3], Stefan Schoenfelder [2,3,7], Anne E. Corcoran [1,2,7] & Peter J. Rugg-Gunn [3,5,7 ✉]

The transition from naive to primed pluripotency is accompanied by an extensive reorganisation of transcriptional and epigenetic programmes. However, the role of transcriptional enhancers and three-dimensional chromatin organisation in coordinating these developmental programmes remains incompletely understood. Here, we generate a high-resolution atlas of gene regulatory interactions, chromatin profiles and transcription factor occupancy in naive and primed human pluripotent stem cells, and develop a network-graph approach to examine the atlas at multiple spatial scales. We uncover highly connected promoter hubs that change substantially in interaction frequency and in transcriptional co-regulation between pluripotent states. Small hubs frequently merge to form larger networks in primed cells, often linked by newly-formed Polycomb-associated interactions. We identify widespread state-specific differences in enhancer activity and interactivity that correspond with an extensive reconfiguration of OCT4, SOX2 and NANOG binding and target gene expression. These findings provide multilayered insights into the chromatin-based gene regulatory control of human pluripotent states.

---

[1] Lymphocyte Signalling and Development Programme, Babraham Institute, Cambridge, UK. [2] Nuclear Dynamics Programme, Babraham Institute, Cambridge, UK. [3] Epigenetics Programme, Babraham Institute, Cambridge, UK. [4] Centre for Computational Biology, University of Birmingham, Birmingham, UK. [5] Wellcome – MRC Cambridge Stem Cell Institute, Cambridge, UK. [6] These authors contributed equally: Peter Chovanec, Amanda J. Collier. [7] These authors jointly supervised this work: Stefan Schoenfelder, Anne E. Corcoran, Peter J. Rugg-Gunn. ✉email: peter.rugg-gunn@babraham.ac.uk

The spatial organisation of chromosomes instructs dynamic processes during development that underpin genome regulation and transcriptional control[1–5]. Mammalian genomes are compartmentalised into a nested hierarchy of structural features[6,7], including megabase-scale topologically associating domains (TADs) that are largely invariant across cell types, and smaller, nested sub-TADs, chromatin loops and insulating neighbourhoods that are frequently reorganised between cellular states[8–17]. Constrained within the larger structural features, distal regulatory elements, such as enhancers, interact with target gene promoters through DNA looping to control promoter activity. Distal elements can loop to several genes, and genes can interact with multiple distal elements, resulting in complex three-dimensional (3D) interaction networks that are often re-wired upon cell state change[5,14,18,19]. The formation of regulatory interactions is directed in part by the cell type-specific occupancy of chromatin proteins and transcription factors[20]. The binding of these factors is often sensitive to epigenetic marks such as DNA methylation, and thus the close interplay between epigenetics and chromatin topology is critical for the appropriate gene regulatory control of cell state transitions. Investigating how chromatin interactions track with a changing epigenetic and gene regulatory landscape is important for understanding the principles of genome organisation and transcriptional control.

One of the most striking periods of epigenome reorganisation occurs as pluripotent stem cells (PSCs) transition from a naive state to a primed state. During this transition, DNA methylation levels rise from ~20% to ~70% genome-wide and there is a dramatic gain in promoter-associated H3K27me3 at several thousand genes[21–27]. These events recapitulate similar molecular transitions as embryos progress from pre- to post-implantation stages of development[28–31] and PSCs therefore provide a tractable system to investigate these processes. In mouse PSCs, global epigenetic changes occur in parallel with the reorganisation of gene regulatory interactions that take place within largely preserved structural domains[17,32]. For example, the acquisition of H3K27me3 at gene promoters during the transition from a naive to a primed state is concomitant with the emergence of a long-range network of promoter–promoter interactions that connect the H3K27me3-marked regions[33]. This spatial network is thought to constrain the transcriptional activity of developmental genes[34]. In human PSCs, the mapping of a subset of DNA interactions that are cohesin-associated revealed that the positioning of long-range structural loops is similar between naive and primed states, suggesting that their TAD and insulated neighbourhood structures are largely preserved[35]. Within those domains, however, individual interactions connecting active enhancers to cell-type-specific genes are re-wired between naive and primed states. This raises the possibility that nested within TADs and cohesin-mediated loops, altered interactions between genes and their regulatory elements have widespread effects on controlling transcriptional changes in pluripotent state transitions. There is a pressing need, therefore, to compare regulatory interactions between enhancers and their target genes in human PSC states globally and at high resolution, and to use these data to unravel the nested structures of genome organisation.

Chromatin conformation capture technologies such as Hi-C can reveal genome-wide spatial DNA interactions[36] and can be combined with sequence enrichment to increase coverage of interactions at genomic features such as promoters (promoter-capture Hi-C; PCHi-C)[37,38]. The rapid progress in experimental methods requires new ways to maximise the discovery of novel insights from the resulting interaction data[39]. Network approaches have been applied successfully to global Hi-C data by using network modularity to identify communities of interacting loci that represent high-level chromatin features such as TADs and

sub-TADs[40–42]. In particular, Pancaldi and colleagues pioneered the development of chromatin assortativity networks, based on the extent to which promoter-interacting regions share chromatin properties, to uncover the relative importance of specific epigenetic states and transcription factors to gene regulatory organisation[43]. What has not been reported so far, however, is the generation of a combined and highly scalable network to compare the global interactomes and epigenetic landscapes between cell types, nor the application of different visualisation methods to provide additional organisational information to the networks.

To investigate the hierarchical differences in 3D genome organisation between naive and primed human PSCs, we used Hi-C[36], PCHi-C[37,38], and chromatin profiling to generate an annotated high-resolution atlas of ~130,000 DNA interactions in these two cell states. By developing a computational approach to integrate and visualise the atlas at a network level, we identify striking differences in organisational features between the two cell states at multiple spatial scales. Prominent genomic features included large, highly connected hubs of interacting genes, such as the proto-cadherin and histone H1 families, which changed substantially in interaction frequency and in their transcriptional co-regulation between pluripotent states. Additionally, small hubs frequently merged through newly formed Polycomb-associated interactions to form larger networks in primed cells. We also identified widespread state-specific differences in enhancer activity and interactivity, whereby a surprisingly low proportion of active enhancers were shared between the two pluripotent cell types. By mapping pluripotency factor occupancy, we found that changes in enhancer state corresponded with an extensive reconfiguration of OCT4, SOX2 and NANOG binding and transcription of their target gene promoters. Our findings uncover a global reorganisation of promoter interactions at multiple spatial scales that occurs during the transition from naive to primed PSCs, thereby providing insights into the chromatin-based gene regulatory control of human development and pluripotency.

## Results

**Mapping promoter interactions and chromatin states in naive and primed human PSCs.** We used Hi-C and PCHi-C to profile the global, high-resolution interactomes of 22,101 promoters in isogenic human naive and primed PSCs (cultured in t2iL+PKCi[22] and TeSR-E8[44] feeder-free conditions, respectively). There was a strong concordance in pairwise interaction read counts between the biological replicates of the same cell type ($r^2 > 0.95$; Supplementary Fig. 1a). PCHi-C data normalisation and signal detection using the CHiCAGO pipeline[37,45] identified 75,091 significant cis-interactions between baited promoters and other genomic regions in naive PSCs, and 83,782 in primed PSCs (Supplementary Fig. 1b). Just under half of the interactions were common to both cell types ($n = 39,360$). Pile-up plots showed that interaction signals are present in the same regions in both the PCHi-C and Hi-C datasets, and, as anticipated from prior studies[37,38,45], the signals are stronger in the PCHi-C data due to the higher coverage at these interacting regions compared with their respective Hi-C samples (Supplementary Fig. 1c). As expected, trans-interactions represented a small minority of promoter interactions (354 interactions). In both cell types, the majority of significant interactions were between the promoters of protein-coding genes and non-promoter genomic regions (Supplementary Fig. 1b). To functionally annotate the interactomes, we integrated new and published histone ChIP-seq datasets to assign ChromHMM-defined[46] chromatin states to the genome including interacting regions (Supplementary Fig. 1d, e). These chromatin state maps identified regions that are associated with active and poised classes of promoters and enhancers, Polycomb-bound sites and

heterochromatin marks. Processed datasets from this large-scale resource are available through the Open Science Framework (https://osf.io/jp29m) and Supplementary Dataset 1, and sequencing data have been deposited to the Gene Expression Omnibus (accession 'GSE133126').

**Network visualisation of promoter interactomes**. We next developed a computational approach called Canvas (Chromosome architecture network visualisation at scales) to visualise and integrate high-resolution, capture-based DNA interaction data and chromatin states at a network scale. Network graphs were constructed where each node of the network represents an individual *Hin*dIII genomic fragment (average size, 4 kb) and each edge represents a CHiCAGO-called[45] significant interaction between nodes (Fig. 1a). We combined all significant interactions detected in naive and primed PSCs to produce a single, unified network graph, which retains information about whether an interaction is shared or cell type-specific (Fig. 1a). The combined network was visualised with a force-directed layout[47] that positions highly interacting nodes closer together and pulls less interacting nodes apart (Fig. 1b).

Annotating naive-specific and primed-specific interactions onto the combined network uncovered large clusters with high and uniform cell-type-specific interactivity, which have not been reported previously in PSCs. Two of the most prominent examples of this are the histone *H1* genes in naive PSCs and the protocadherin genes in primed PSCs (Fig. 1b and Supplementary Fig. 2a). The histone *H1* cluster contained 198 nodes connected by 984 edges, and the protocadherin cluster contained 165 nodes and 1188 edges. Overlaying transcriptional information onto each node showed that the higher promoter interactivity within the histone *H1* and in the protocadherin clusters is associated with increased expression of nearly all genes within each cluster, implying a coordinated transcriptional response (Fig. 1c and Supplementary Fig. 2b). Not all large gene clusters that changed interaction frequency between cell types showed differential expression, as exemplified by the keratin and olfactory receptor gene regions (Fig. 1b and Supplementary Fig. 2a, b). We also generated network graphs using each individual replicate to assess the reproducibility of the networks. We found that the key interaction landmarks are very consistent when comparing datasets, demonstrating good reproducibility between replicates (Supplementary Fig. 2c, d). Taken together, the combination of high-resolution DNA interactivity and gene regulatory information on a whole-network graph provides a systems-level visualisation of genome organisation, and uncovered large clusters of promoter interactions as prominent structural features in pluripotent cells.

**Multiscale exploration of promoter-interaction networks**. Network reconstruction of the promoter-interaction data using Canvas allows the interrogation of genome organisation at multiple spatial levels. Viewed at the lowest magnification, the network graph consists of >3000 individual sub-networks of varying size (Fig. 1d). Using algorithms developed for the detection of community structure[48], each sub-network can be further divided into communities that have high internal interactivity. We noticed that the positioning of the individual communities showed strong, significant overlap with TADs, which we identified separately using matched Hi-C data (Supplementary Fig. 3a–d). This finding demonstrates that Canvas is capable of using promoter-interaction data to define and visualise structural components of genome organisation. In agreement with previous studies[35,49], the majority (70%) of TADs are shared between naive and primed PSCs (Supplementary Fig. 3a).

We additionally found, however, that the insulation score of TAD boundaries was higher in primed compared to naive PSCs (by 15%; naive: 0.80, primed 0.92; Supplementary Fig. 3b), suggesting there are differences in TAD boundary strength between the two cell types. In addition to communities, this mid-level visualisation of the network graph identified large interaction clusters, exemplified by the histone H1 and protocadherin genes that we described above. Finally, the highest visualisation level identifies high-resolution and detailed structure including potential *cis*-regulatory regions of individual baited gene promoters (Fig. 1d). Taken together, network scale visualisation of annotated promoter interactomes can provide an intuitive method for data exploration at several scales of 3D chromosome organisation to reveal features of genome architecture that range from higher-order chromatin structure down to individual promoter interactions.

**Long-range promoter interactions distinguish the two pluripotent states**. To investigate the spatial regulation of human pluripotent states, we examined promoter-interaction changes that occur within each defined sub-network between naive and primed PSCs. We compared the number of nodes and the overall number of interactions within individual sub-networks. Using this approach revealed that nearly all of the sub-networks that changed in the number of nodes and interactions between pluripotent states were larger in primed PSCs (Fig. 2a). Sub-networks in primed PSCs therefore contained a greater number of interactions and more nodes compared to sub-networks in naive PSCs (Fig. 2a). We observed similar trends when examining each replicate separately (Supplementary Fig. 4a). These findings suggest that smaller communities in naive PSCs come together to form larger sub-networks in primed PSCs.

To begin to understand the increased size of sub-networks in primed PSCs, we examined the sub-network that showed the largest change in the number of nodes and edges when comparing between pluripotent states. In naive PSCs, this sub-network was composed mainly of individual, separate communities that are distributed across chromosome 5 (Fig. 2b). However, in primed PSCs, these communities were connected by >200 long-range interactions (defined as >1 Mb in the linear distance) (Fig. 2b). The acquisition of long-range interactions to create large sub-networks in primed PSCs was a common feature, as exemplified by regions on other chromosomes including the *HOXA*, *HOXD* and *NKX* loci (Supplementary Fig. 4b, c). In contrast, the only clear instance of a sub-network with more promoter interactions and nodes in naive PSCs was for the histone H1 locus (Supplementary Fig. 4b, c). Analysing all promoter interactions revealed a substantial increase in their number and linear distance in primed compared to naive PSCs (Fig. 2c) and this difference was independent of the applied CHiCAGO threshold (Supplementary Fig. 4d).

We next confirmed the difference in the number of long-range promoter interactions between pluripotent states using an alternative approach that is independent of promoter capture. We analysed our Hi-C data using HiCCUPS[12], which is an algorithm that calls interaction 'peaks' when a pair of loci show elevated contact frequency relative to the local background. This approach revealed that the number of long-range chromatin interactions (>1 Mb) was substantially higher in primed ($n = 889$) compared to naive PSCs ($n = 480$) (Fig. 2d). This striking difference is exemplified for several chromosomes by overlaying the identified peaks onto Hi-C contact matrices (Fig. 2e and Supplementary Fig. 5). Importantly, the interaction peaks identified by HiCCUPS in primed PSCs matched the positions of long-range promoter interactions detected by PCHi-C (Fig. 2e),

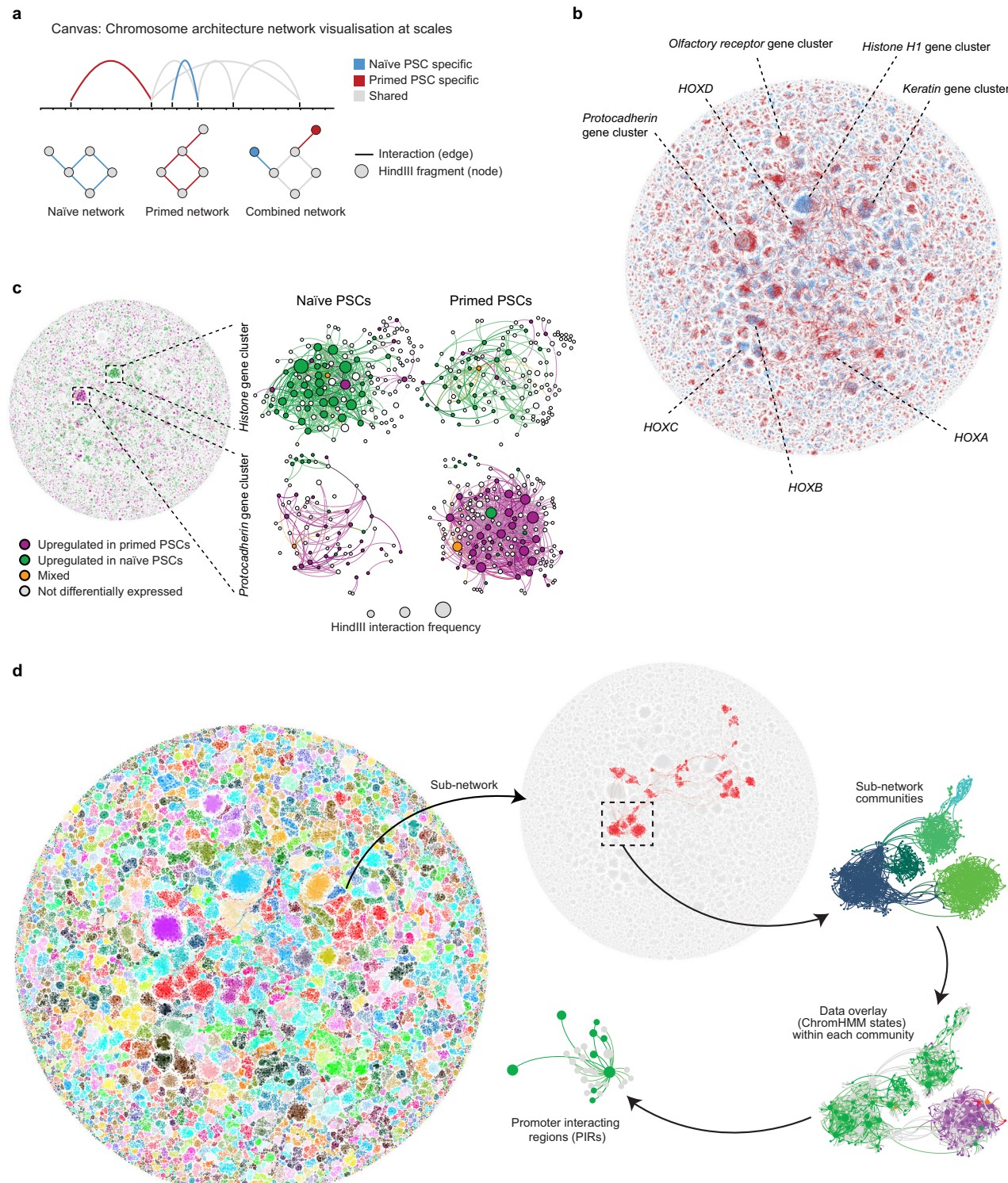

thereby validating the presence of the promoter-capture interactions. To visualise the contact enrichment at the identified peaks in naive and primed PSCs more closely, we applied Aggregate Peak Analysis[12] to the long-range interaction loci. We found there was a stronger enrichment (by ~2.5-fold) at these aggregated sites over local background levels in primed PSCs compared to naive PSCs (Fig. 2f). This result confirms the observed differences in long-range contact frequency between pluripotent cell types. Taken together, long-range promoter interactions are a dominant feature in primed PSCs that connect individual communities into larger sub-networks.

## Acquisition of Polycomb-associated interaction networks in primed human PSCs. To characterise the properties of the long-range promoter interactions, we investigated the chromatin states of the individual *Hin*dIII fragments. The chromatin state of regions that are brought together by long-range interactions

**Fig. 1 Multiscale exploration of promoter-interaction data using force-directed network graphs. a** Representation of PCHi-C data as arc diagrams (upper) and as corresponding network graphs (lower). Interacting HindIII genomic fragments are depicted as nodes that are connected by edges (significant interactions). A combined network graph is created by merging naive and primed human PSC datasets whilst retaining cell type-specific information. Blue, naive-specific nodes and edges; red, primed-specific nodes and edges; grey, shared nodes and edges. **b** Canvas produces a force-directed layout of the combined, whole-network graph. Nodes that interact more frequently are pulled closer together, and less interacting nodes are pushed further apart. **c** Differential gene expression (*P*-adj < 0.05; *n* = 3 biologically independent samples per cell type) categories between naive and primed PSCs overlaid onto the combined network graph. The 'mixed' category refers to the small subset of nodes that contain two or more genes that differ in the direction of their transcriptional change. Expanded examples are shown for the histone *H1* gene cluster (upper) and the protocadherin gene cluster (lower). The size of each node corresponds to the number of interactions (degree) of the node. **d** The network graph (left) shows the combined promoter interactomes for naive and primed PSCs, visualised using Canvas. Each sub-network is assigned a different colour. Following the arrow, the next part of the figure shows a single, isolated sub-network (the largest sub-network in the dataset). The boxed region is expanded in the next area of the figure to show the distinct, sub-network communities that have high internal interactivity. The same communities and structure are shown in the next part of the figure, where the nodes are coloured according to their predominant chromatin state (green, active; orange, bivalent; red, polycomb-only; purple, heterochromatin; grey, background). A small part of the community is magnified in the final area of the figure to show all of the interacting regions for one promoter.

differed between the two cell types. The longest interactions in naive PSCs were marked by active chromatin states at the promoter and at the interacting region (Fig. 3a, left). The clear majority of these interactions and their chromatin states were also present in primed PSCs (Fig. 3a, left). In contrast, the longest interactions in primed PSCs were dominated by bivalent chromatin states (dual H3K27me3 and H3K4me1/3 methylation) at both ends of the interaction (Fig. 3a, right). Only one-third of these regions were classified as bivalent in naive compared to primed PSCs (208/1000, naive; 655/1000, primed) and very few of these interactions were detected in naive PSCs (*n* = 37/1000; Fig. 3a), suggesting that they are formed de novo upon the transition from naive to primed pluripotency.

To further investigate the establishment of H3K27me3-associated interaction networks in primed PSCs, we examined the chromatin state of these same regions in naive PSCs. Approximately one-quarter of the regions were already marked by H3K27me3 in naive PSCs (Fig. 3b). More commonly, the regions were classified as active or mixed state chromatin in naive PSCs, and therefore acquired H3K27me3 during the transition to primed PSCs (Fig. 3b). A closer look at the individual HindIII fragments that were classified as mixed chromatin state in naive PSCs revealed that these regions contained patches of active and repressive chromatin states (Supplementary Fig. 6a), commonly with H3K4me1/3-only peaks residing within larger blocks of H3K27me3. This implies that the H3K4me1/3-marked sites are protected from H3K27me3 in naive PSCs, but that H3K27me3 spreads throughout the region in primed PSCs. Genes within these regions were associated with developmental processes, with examples including *DLX*, *GATA* and *HOX* factors (Supplementary Fig. 6b).

Using Canvas to visualise all H3K27me3-associated interactions clearly highlights the differences between pluripotent states. This category of interactions formed numerous highly interacting clusters in primed but not naive PSCs (Fig. 3c). Individual clusters were connected through long-range interactions and, remarkably, nearly all (98%) of the long-range *cis*-interactions within the dataset were associated with bivalently-marked promoters (Supplementary Fig. 6c).

Protein-coding genes belonging to the major developmental gene families were spatially organised within the H3K27me3-associated interaction network in primed PSCs (*n* = 696; Fig. 3c). This large gene set was strongly enriched for transcriptional regulators and homeobox-containing factors (Supplementary Dataset 2). For example, a region on chromosome 7 that includes *SHH*, *EN2* and *MNX1* formed a highly interacting cluster in primed PSCs through the presence of long-range interactions that align closely to H3K27me3 peaks (Fig. 3d). Other examples include the *HOX* gene loci, where we detected long-range

cis-interactions and also *trans*-interactions with regions on other chromosomes, including the *HOX* clusters themselves (Supplementary Fig. 7a). The lower levels of H3K27me3 at gene promoters in naive PSCs corresponded to the absence of interactions within the *HOX* loci (Supplementary Fig. 7b, c). Three-dimensional DNA-FISH experiments validated these findings by showing that in primed PSCs, *HOXD10/11* and *DLX1/2* loci were in closer proximity compared to *HOXD10/11* and a control locus that is equidistant in the opposite direction along the chromosome (Supplementary Fig. 7d). In contrast, in naive PSCs, there was no difference in the proximity between *HOXD10/11*—*DLX1/2* and *HOXD10/11*—control locus (Supplementary Fig. 7d). Taken together, these results reveal that the majority of long-range interactions connect regions that gain H3K27me3 during the naive to primed conversion, thereby creating large spatial networks of developmental genes in primed PSCs.

**Pluripotent state-specific enhancer activity and interactivity.** Our current understanding of the differences in gene regulatory control between human pluripotent states is incomplete due to the lack of global and high-resolution mapping of promoter cis-regulatory interactions. To overcome this important knowledge gap, we annotated enhancers in naive and primed PSCs and then used PCHi-C data to identify the target gene promoters for those enhancers. We defined super-enhancers (SEs) by running H3K27ac ChIP-seq data through the ranking of super-enhancer (ROSE) pipeline[50,51]. This approach identified 182 naive-specific SEs and 62 primed-specific SEs (Fig. 4a). We also curated ~600–700 SEs that are shared between both cell types (Fig. 4a). Integrating the enhancer annotations with the PCHi-C data identified the gene promoters that interact with each SE. Remarkably, the majority of SE-target genes (85%, n=931) were cell type-specific and only 15% of genes were contacted by SEs in both naive and primed PSCs (Fig. 4b). In particular, there were a large number of gene promoters (n=633) that interacted with a shared SE only in naive PSCs (Supplementary Fig. 8a). In contrast, relatively few promoters (n=250) interacted with a shared SE only in primed PSCs (Supplementary Fig. 8a). Based on this, we hypothesised that SEs might interact with more promoters in naive compared to primed PSCs, however, we found that the number of interacting promoters per SE was very similar between the two cell types (Supplementary Fig. 8b,c). Genes that interacted with SEs only in naive PSCs included members of signalling pathways such as *IL6* and *GDF3* and chromatin regulators such as *TET1* and *REST* (Fig. 4b). Similarly, genes in contact with a SE only in primed PSCs included transcription factors such as *KLF7*, *TCF4* and *ZIC2* (Fig. 4b). Relatively few genes (*n* = 170) interacted with a SE in both cell types and this set of genes included

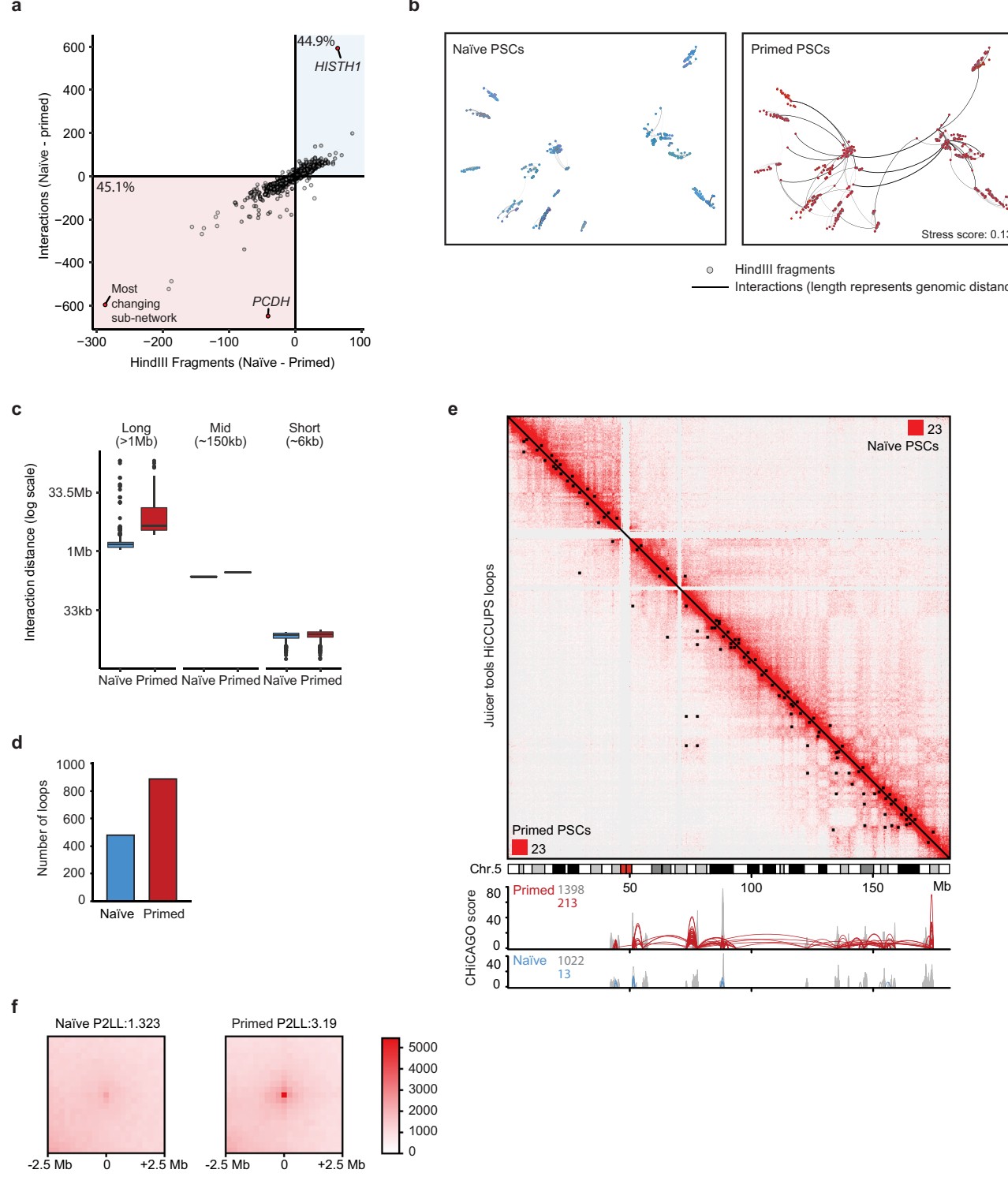

ACTB and LIN28B (Fig. 4b). In line with the proposed capability of SEs to promote the expression of genes that are important for cell identity[50,52], we observed a higher transcriptional output of SE-target genes in a cell type-specific manner and a comparable, high transcriptional output for genes contacted by SEs in both cell types (Fig. 4c). The large number of cell type-dependent gene promoters that contact SEs demonstrates the substantial changes in SE activity and interactivity that occur between the two pluripotent states.

We next examined interactions between promoters and enhancer elements other than SEs (Fig. 4d). Of the gene promoters detected as interacting with a normal enhancer, 49% were found to contact an enhancer in both cell types (Fig. 4e and Supplementary Fig. 8d), which contrasts with the much lower proportion for SEs (15%; Fig. 4b). Several core pluripotency-associated genes interact with enhancers in both naive and primed PSCs including DPPA4, TCF3 and TFAP2C—all of which are highly expressed in both cell types. More generally, differences in enhancer interactivity were concordant with transcriptional changes of their target genes in naive and primed PSCs (Fig. 4f). Genes that interact with enhancers only in naive PSCs include RRAD and YPEL2, and in primed PSCs this gene set includes

**Fig. 2 Long-range promoter interactions in primed PSCs drive genome conformation changes between pluripotent states. a** Plot shows the number of interactions (edges) and the number of interacting *Hind*III fragments (nodes) for each sub-network in naive and primed PSCs. Each small circle represents a different, individual sub-network. The lower-left quadrant reveals sub-networks that are larger in primed PSCs, and the upper-right quadrant shows the small number of sub-networks that are larger in naive PSCs. The protocadherin (*PCDH*), histone *H1* (*HISTH1*) and 'most changing' (containing diverse genes) sub-networks are highlighted in red. **b** Multidimensional scaling representation (MDS) of the 'most changing' sub-network plotted using the linear genomic distance between nodes as edge weights. The measured stress score of 0.139 indicates there is a reasonable fit between the linear genomic distances and the spacing of the nodes as determined by MDS[111]. **c** Plot shows the distribution of linear genomic distances between interacting nodes in naive and primed PSCs, binned into long-, mid-, and short-range distances ($n = 1000$). The box bounds the interquartile range (IQR) divided by the median (horizontal line), and Tukey-style whiskers extend to a maximum of 1.5 × IQR beyond the box. Circles indicate individual outliers. **d** Plot shows the total number of long-range chromatin interactions identified on all chromosomes in naive and primed PSCs. **e** Hi-C interaction matrix of chromosome 5 at a resolution of 250 kb with Knight–Ruiz (KR) normalisation; upper right, naive PSCs; lower left, primed PSCs. Areas of contact enrichment were defined separately for naive and primed PSCs using HiCCUPS and each cell-type-specific set of chromatin interactions are highlighted as a black square on their respective heatmap. The two corner numbers indicate the maximum intensity values for the matrix. The tracks below the Hi-C heatmap show the PCHi-C interactions and CHiCAGO scores over the same region. **f** Heatmap shows the aggregate peak analysis (250 kb resolution) of all naive and primed chromatin interactions genome-wide for naive and primed PSCs. Chromatin interactions >5 Mb from the diagonal were used for the analysis. The 'peak to lower left' (P2LL) score denotes the enrichment of the central pixel over the pixels in the lower left quadrant.

*DUSP6* and *OTX2*. Given the differences in the transcriptional activity and regulatory control of the identified genes, this integrated dataset uncovers factors that could have important cell type-specific functions.

The resultant dataset uncovered changes in promoter–enhancer interactions that occur between naive and primed PSCs, thereby revealing insights into gene regulatory control of human pluripotency. For example, *DPPA5* is highly transcribed in naive PSCs and the promoter interacts with SEs that are marked by high levels of H3K27ac and H3K4me1 (Fig. 4g). In contrast, in primed PSCs, this region has strongly reduced H3K27ac levels, there are no SEs or detectable *DPPA5* promoter interactions, and *DPPA5* is not transcribed (Fig. 4g). A second example shown is *TBX3*, which is more highly transcribed in naive compared to primed PSCs, and this corresponds to the presence of *TBX3* promoter interactions with enhancers only in naive PSCs (Supplementary Fig. 8e). We found that this locus switched from predominantly active chromatin marks in naive PSCs to high levels of H3K27me3 in primed PSCs (Supplementary Fig. 8e). In primed PSCs, the *TBX3* promoter formed long-range interactions with several other H3K27me3-marked sites including *TBX5* and *LHX5* (Supplementary Fig. 8e).

We next took advantage of Canvas to examine promoter–enhancer communication on a network scale (Fig. 4h). The global visualisation of promoter interactions and enhancer annotations revealed clusters of co-regulated genes with many hubs containing both SEs and enhancers (Fig. 4h). Collectively, our global and high-resolution mapping of promoter cis-regulatory interactions revealed that most SE-target gene contacts were cell type-specific, identifying close to 1000 genes with SE interactions that differed between naive and primed PSCs. Importantly, these integrated datasets provide a valuable resource to investigate the changes in the activity and target gene interactivity of putative regulatory regions in human pluripotent states.

**Decommissioning of naive-specific active enhancers correlates with the acquisition of DNA methylation**. To better understand the changes in enhancer activity between human pluripotent states, we next examined how naive-specific active enhancers could be decommissioned in primed PSCs. By far the most common change at these sites was an increase in DNA methylation levels from an average of ~25% in naive PSCs to ~90% in primed PSCs (Fig. 5a). The gain in DNA methylation at these regions was very similar to the increase that occurs genome-wide, as indicated by the 'background' category and reported previously (Fig. 5a [22,53]). In naive PSCs, all enhancer categories were, overall, less methylated compared to background (>10% difference in

median DNA methylation levels). In primed PSCs, however, only primed-specific and shared enhancer categories showed lower DNA methylation than background (by >10%), suggesting that these regions are protected to some extent from global events (Fig. 5a). Examination of histone ChIP-seq data confirmed the expected reduction in H3K27ac and H3K4me1 levels at naive-specific active enhancers when comparing between naive and primed PSCs (Fig. 5b, log2 fold-change of median >1). Overall, naive-specific active enhancers did not gain H3K9me3 or H3K27me3 in primed PSCs (log2 fold-change of median <1), thereby demonstrating that the acquisition of repressive histone modifications is not a common mode of enhancer decommissioning in primed PSCs (Fig. 5b). However, we identified a set of 75 enhancers that acquired H3K27me3 in primed PSCs and these regions were associated with developmental genes including *GATA3*, *TFAP2A*, *NEUROG1* and *TBX3* (Fig. 5c and Supplementary Fig. 8e). In keeping with the common anticorrelation between H3K27me3 and DNA methylation marks[54], most of the enhancers that gained H3K27me3 remained DNA hypomethylated in primed PSCs (Fig. 5d). Thus, the majority of naive-specific active enhancers are likely to be decommissioned by acquiring DNA methylation, however, a small subset of these enhancers adopt a Polycomb-associated chromatin state in primed PSCs. Taken together, this integrated data resource provides a large collection of putative regulatory sequences and their target promoters in naive and primed PSCs, underpinned by network interactions, thereby revealing the dynamic changes in gene regulatory control between human pluripotent states.

**Widespread reorganisation in OCT4, SOX2 and NANOG occupancy between human pluripotent states**. To begin to understand how the dynamic regulation of enhancers is controlled between human pluripotent states, we investigated the association between transcription factor binding and enhancer activation. After integrating new and previously generated ChIP-seq data, we observed that the shared pluripotency factors OCT4, SOX2 and NANOG (OSN) showed cell type-specific binding at *DPPA5* promoter-interacting regions, whereby OSN was bound at the interacting SEs in naive PSCs (Fig. 4g). To study this association at a genome-wide level, we first asked whether OSN-bound sites overlap with particular chromatin state categories. The results showed that OSN occupancy is strongly associated with active enhancers in both cell types (Fig. 6a). Essentially all OSN sites in both cell types contained enhancer chromatin signatures including H3K27ac (92% with log2 RPM > 0) and H3K4me1 signals (98%) and open ATAC-seq regions (84%, Supplementary Fig. 9a). A small subset of OSN regions was also positive for H3K4me3 (12%)

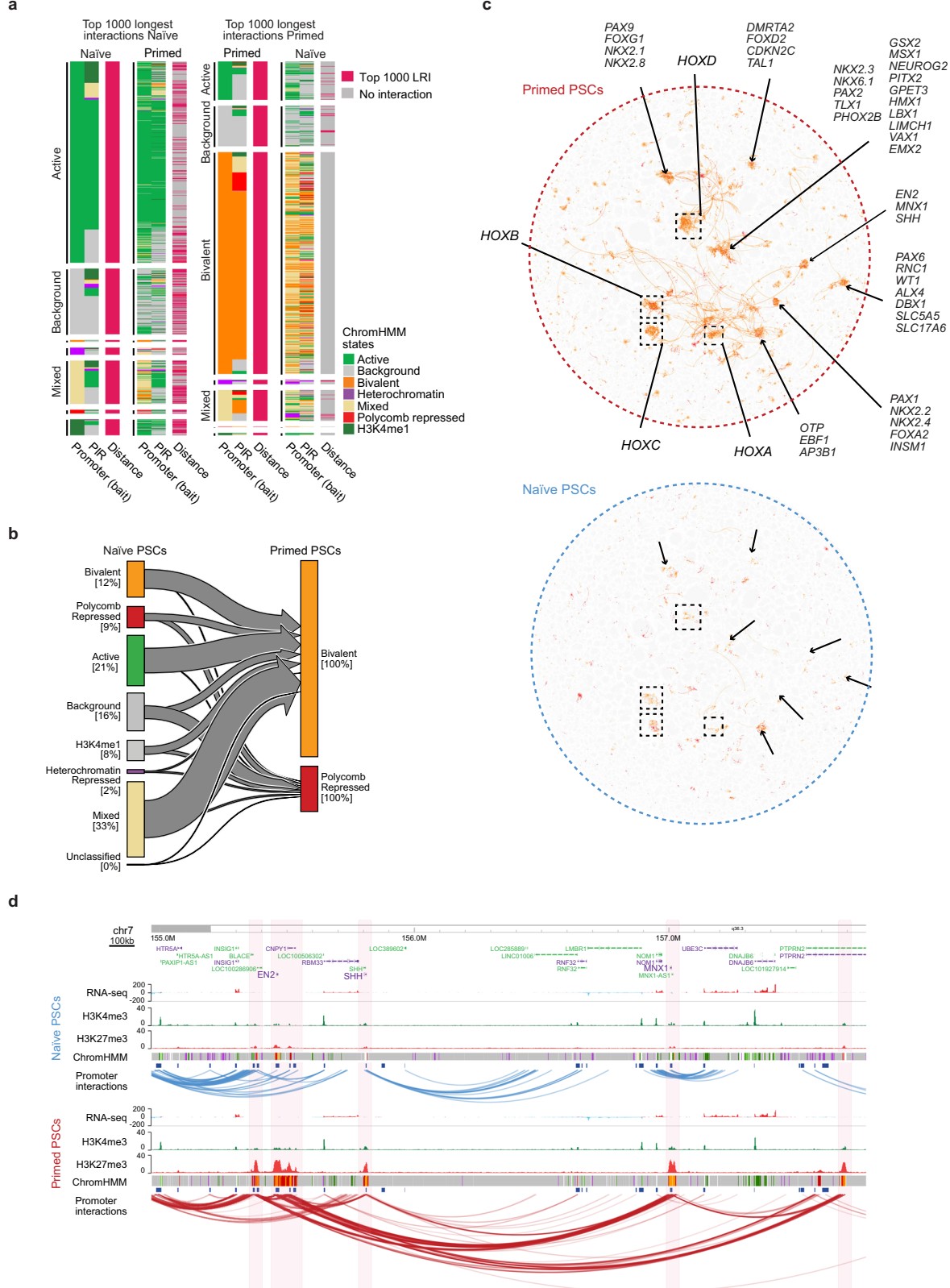

while showing lower H3K4me1 (Supplementary Fig. 9a), suggesting that they have promoter activity. While still included in our OSN peak annotation, these regions have a comparatively lower OSN signal (Supplementary Fig. 9a). Consistently, distal OSN sites had higher OSN signal than those overlapping transcriptional start sites (Supplementary Fig. 9b), reinforcing the link

between OSN binding and enhancer activity. Similarly, nearly all active enhancers showed OSN signal (Fig. 6b, c). These results demonstrate a remarkable overlap between OSN occupancy and active enhancers in naive and primed PSCs.

Despite the strong associations between OSN binding and active enhancers in both cell types, we found that only

**Fig. 3 Long-range promoter interactions are associated with bivalent chromatin in primed human PSCs. a** Heatmaps show the chromatin states of the promoters and promoter-interacting regions (PIRs) that are connected by the longest interactions in naive (left) and primed (right) PSCs. Each heatmap is ordered and divided by the chromatin state of the promoter *Hind*III fragment. **b** Plot shows the chromatin states of *Hind*III fragments in naive PSCs that transition into the bivalent or Polycomb-associated states in primed PSCs. The percentages shown are within each state. **c** Visualising all bivalent and Polycomb-associated interactions on the network graph highlights several interaction clusters particularly in primed PSCs (upper) that contain many developmental genes. The clusters, together with the interactions, are largely absent in naive PSCs (lower). **d** Genome browser view in naive (upper) and primed (lower) PSCs of a ~2.5 Mb region that contains several developmental genes including *EN2*, *SHH* and *MNX1*. Significant promoter interactions are shown as blue and red arcs. ChIP-seq (H3K4me3 and H3K27me3) and strand-specific RNA-seq tracks are shown. Chromatin states for each genomic region were defined by ChromHMM[46] using ChIP-seq data (active chromatin, light green; H3K4me1-only chromatin, dark green; bivalent chromatin, orange; Polycomb-associated, red; heterochromatin, purple; background, grey).

one-quarter of active enhancers are shared between naive and primed PSCs (Supplementary Dataset 3). This unexpected observation suggests there is a widespread remodelling of OSN occupancy and enhancer activity between naive and primed PSCs. Furthermore, when OSN occupancy was lost during the transition from a naive to a primed state, 74% ($n = 4294$) of all interactions at these sites were also lost, and when OSN was gained, 86% ($n = 247$) of all interactions at these regions were gained (Fig. 6d; $p < 0.0001$; Chi-squared test). Enhancers that either retained OSN binding or were not bound by OSN typically retained or gained interactions (Fig. 6d), which is probably due to the higher number of OSN-bound enhancers in naive PSCs (Supplementary Fig. 9a).

Given that OSN binds different active enhancers in each cell type, we reasoned that OSN is likely to be recruited by cell-type-specific factors. As a first look at this, we identified transcription factor motifs that were significantly enriched at OSN sites in one cell type compared to the other. Few primed-specific candidate factors were identified, however, our analysis uncovered several transcription factors that were expressed in naive PSCs and whose binding motif occurred more frequently at OSN sites in naive compared to primed PSCs (Fig. 6e). These factors included KLF5, KLF16, SP transcription factors, TFAP2C and ZFX (Fig. 6e, f). By re-analysing ChIP-seq data[55], we found that the naive-associated transcription factor TFAP2C was enriched (>3-fold) at OSN-bound regions in naive PSCs (Fig. 6g) and indeed ~20% of these regions contained TFAP2C peaks. This percentage is just under the proportion of OSN-occupied sites in naive PSC that contain a TFAP2C motif (~30%; Fig. 6f). TFAP2C signal was low at primed-specific OSN sites in naive PSCs, which suggests that TFAP2C does not pre-bind to regions that later acquire OSN occupancy in primed PSCs (Fig. 6g). Taken together, these results lead us to propose that combinations of transcription factors, including TFAP2C but also others, could help to recruit the shared pluripotency factors OSN to active enhancers in naive PSCs. Overall, these findings uncover the widespread and global reorganisation in enhancer activity, interactivity and OSN binding that occurs during the transition between human pluripotent states.

## Discussion

We have generated high-resolution profiles of chromatin interactions and enhancer states in naive and primed PSCs and uncovered widespread rewiring particularly of large interaction sub-networks and also of promoter–enhancer contacts that change between pluripotent cell types (Fig. 7). These findings together with the annotated chromatin interaction maps advance our understanding of the molecular control of gene regulation in pluripotency and in the earliest stages of human development.

We found that interaction sub-networks that are formed of large, highly connected hubs changed substantially in their interaction frequency and, for a subset, also in their transcriptional activity between pluripotent states. A prominent example

of this was a region containing multiple histone *H1* genes, which unexpectedly had higher promoter interactivity and transcriptional output in naive PSCs. This finding is in line with a recent proteomic study that reported a higher abundance of some histone H1 variants in naive compared to primed PSCs[56]. The interaction cluster included sets of histone *H1* genes that are transcribed with cell cycle-dependent and -independent control, which suggests that the difference in the chromatin organisation of this region between pluripotent states is likely to be driven by additional factors that act outside of the cell cycle. Given that the expression and regulation of individual *H1* isoforms vary substantially between cell types and these properties are associated closely with pluripotency, differentiation and development[57,58], a focused examination of histone H1 function in these cell types would be a promising future direction of research. More commonly though, we discovered that sub-networks tended to be larger and to contain a greater number of interactions in primed compared to naive PSCs. Examples of this included the proto-cadherin gene cluster, which encodes cell adhesion molecules that are predominantly expressed in the neural lineage[59]. In neural cells, active protocadherin gene promoters and enhancers are brought together by CTCF and cohesin-mediated DNA loops to form an interaction hub[60]. Our results show that this hub begins to pre-form during early development, potentially priming this region for coordinated activation upon neural development. A recent study reported that during the transition from naive to primed PSCs there is a change in the chromatin signatures of the clustered protocadherin gene locus leading to 'pre-set' patterns of protocadherin gene expression[61]. Together with our findings, these studies imply a coordinated mechanism involving both gene interactions and chromatin states that establishes protocadherin gene control during development. More generally, cell-specific changes in sub-network organisation may also provide opportunities for transcriptional co-regulation and resilience to perturbation. Our study has identified a large cohort of networks that can now be systematically targeted to test these predictions.

The aggregation of interaction hubs into larger networks in primed PSCs was frequently associated with the acquisition of long-range interactions that bridged Polycomb-occupied regions. These events created spatial networks connecting >600 *cis*-regulatory elements that control the transcription of developmental regulators. Similar, smaller-scale, Polycomb-mediated networks have been described in serum-grown mouse PSCs[33,34,62] and the prevalence of Polycomb-associated long-range interactions is strongly reduced after mouse PSCs are transitioned to a naive state[33]. These changes have been attributed in mouse PSCs to the global redistribution of DNA methylation and H3K27me3 that occurs between primed and naive pluripotent states[26,33,63]. H3K4 methyltransferases also orchestrate long-range interactions at enhancers and bivalent promoters[64,65], although whether they show state-specific differences has not been examined. Preventing the redistribution of DNA methylation and H3K27me3 during the transition to a naive state is sufficient to block changes in

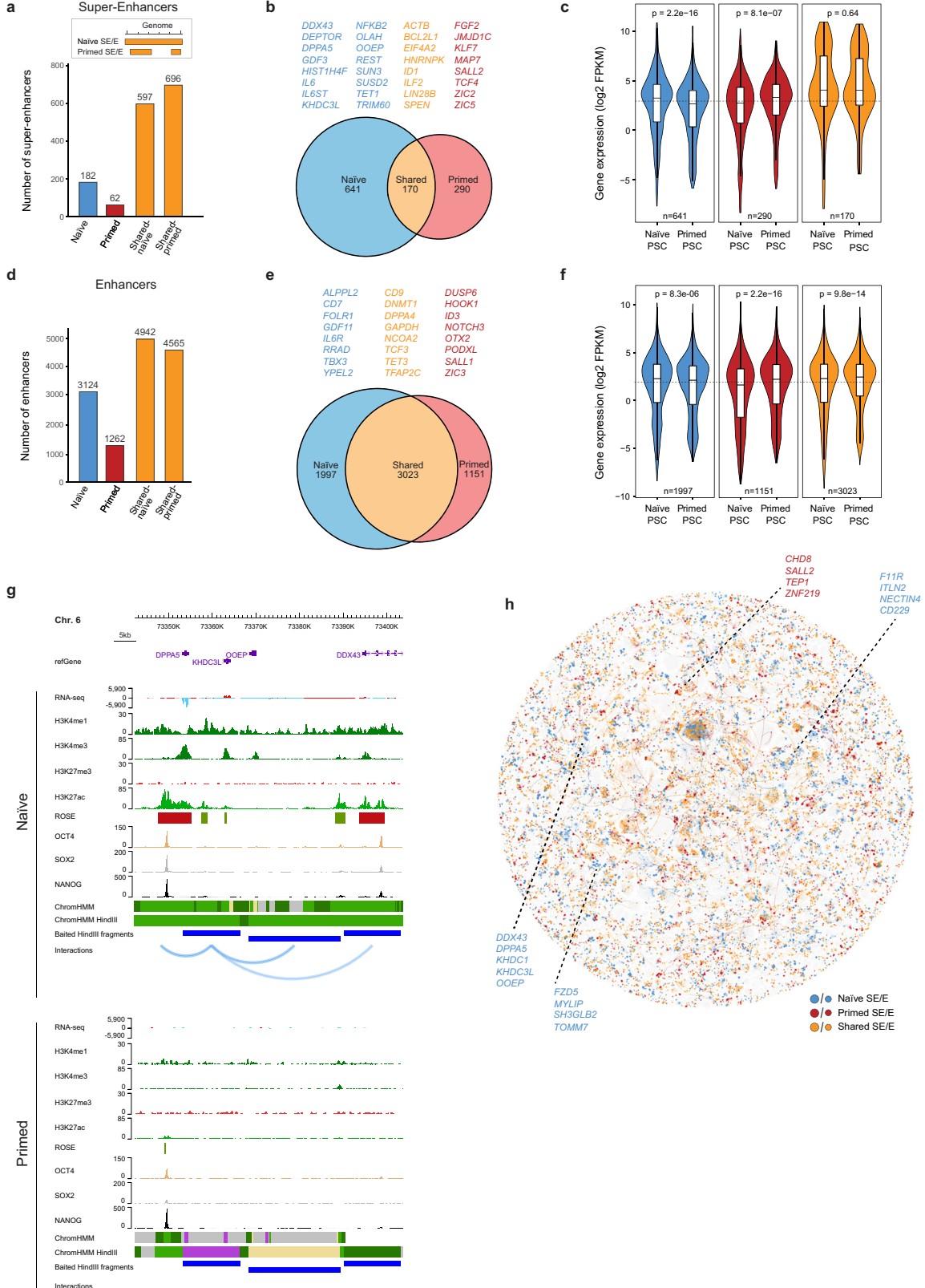

chromatin compaction at several exemplar regions, thereby directly linking epigenome remodelling with aspects of genome organisation[66]. The transcriptome and cell state of mouse and human naive PSCs are largely unaffected by experimentally disrupting Polycomb levels[67–70]. In contrast, primed PSCs are sensitive to the removal of Polycomb proteins[67,70–72]. These

observations collectively imply that the stable transition from a naive to a primed state of pluripotency requires the reconfiguration of DNA interactions to provide a coordinated set of 'poised' regulatory signals to control promoter priming.

A clear difference between human pluripotent cell types that we observed was in enhancer interactivity and activity state, and

**Fig. 4 Dynamics of enhancer activity and interactivity between human pluripotent states. a** Plot showing the number of ROSE-called SEs in naive and primed PSCs. As illustrated in the diagram, two values are given for shared SEs because a SE in one cell type may overlap with two individually called SEs in the other cell type. **b** Diagram showing the number of genes that are contacted by SEs in the two pluripotent cell types. Shared genes (orange) are genes that are contacted by SE elements in both naive and primed PSCs. Naïve-specific genes (blue) and primed-specific genes (red) are contacted by SEs in either naive or primed PSCs, respectively. **c** Plots showing the log2 FPKM expression of genes that interact with SEs in each cell type (naive, $n = 641$ genes; primed, $n = 290$ genes; shared, $n = 170$ genes). The inner box bounds the IQR divided by the median (horizontal line), and Spear-style whiskers extend to the minimum and maximum of the data values. $P$-values are derived from a two-sided Mann–Whitney $U$ test; $n = 3$ biologically independent RNA-seq datasets per cell type. **d** Plot showing the distribution of ROSE-called enhancers in naive and primed PSCs. **e** Diagram showing the number of genes that are contacted by enhancers in the two pluripotent cell types. Genes that are also in contact with a SE have been removed from this list of enhancer-interacting genes. **f** Plots showing the log2 FPKM expression of genes that interact with enhancer elements in each cell type (naive, $n = 1997$ genes; primed, $n = 1151$ genes; shared, $n = 3023$ genes). The inner box bounds the IQR divided by the median (horizontal line), and Spear-style whiskers extend to the minimum and maximum of the data values. $P$-values are derived from a two-sided Mann–Whitney $U$ test; $n = 3$ biologically independent RNA-seq datasets per cell type. **g** Genome browser view of the *DPPA5* promoter interactomes in naive (upper) and primed (lower) PSCs. Significant interactions are shown as blue arcs that connect the baited *Hin*dIII fragment containing the *DPPA5* promoter with promoter-interacting regions. ChIP-seq (H3K4me1, H3K4me3, H3K27me3, H3K27ac, OCT4, SOX2 and NANOG) and strand-specific RNA-seq tracks are shown. Chromatin states include active chromatin, light green; H3K4me1-only chromatin, dark green; bivalent chromatin, purple; background, grey. ROSE tracks show the location of enhancers (green) and super-enhancers (red), and OSN tracks show the position of shared (orange) and naive-specific (blue) regions of OSN occupancy. **h** Network graph showing the locations and cell type-origin of enhancer and SE elements. Colours depict naive-specific (blue), primed-specific (red) and shared (orange) enhancer and SE elements. Node size represents SE (large nodes) and enhancers (small nodes). Lines represent interactions and are coloured according to the colour of the node of origin.

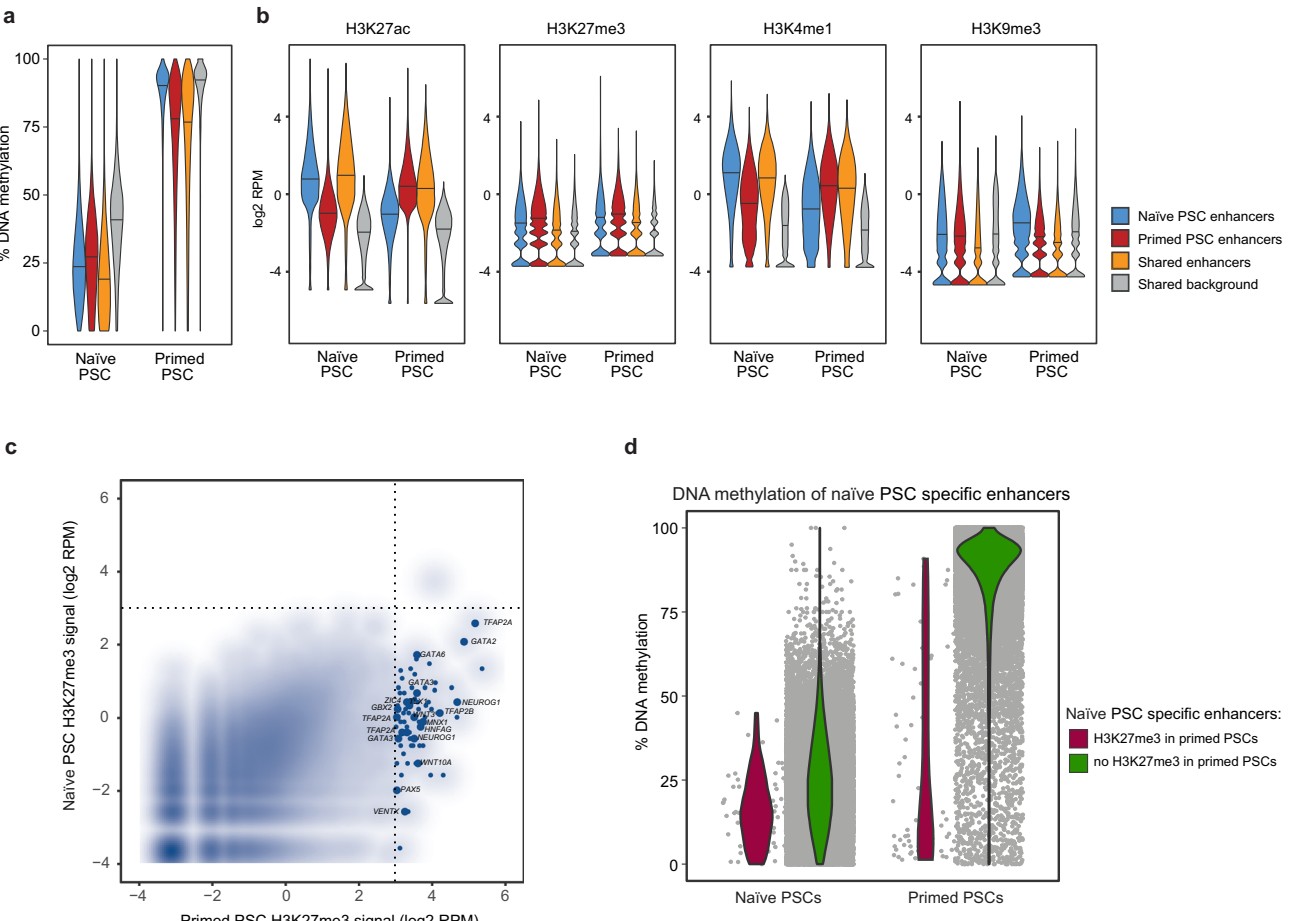

**Fig. 5 Naive-specific active enhancers are decommissioned predominantly by DNA methylation. a**, **b** Plots show the **a** percent DNA methylation or **b** histone modification levels in naive and primed PSCs at shared ($n = 23,371$) and cell type-specific active enhancers (naive, $n = 26,134$; primed, $n = 36,759$). Regions that are in the background chromatin state in both cell types are shown to indicate genome-wide levels ($n = 69,685$). **c** Smoothed scatter plot shows H3K27me3 levels at naive-specific enhancer regions in both pluripotent states. Enhancers that gain H3K27me3 in primed PSCs are highlighted and annotated with their nearest gene. **d** Violin plot shows the per cent DNA methylation of two classes of naive-specific enhancers depending on whether those regions gain (maroon) or do not gain (green) H3K27me3 after the transition to a primed state. Naive-specific enhancers that acquire H3K27me3 are protected from DNA hypermethylation.

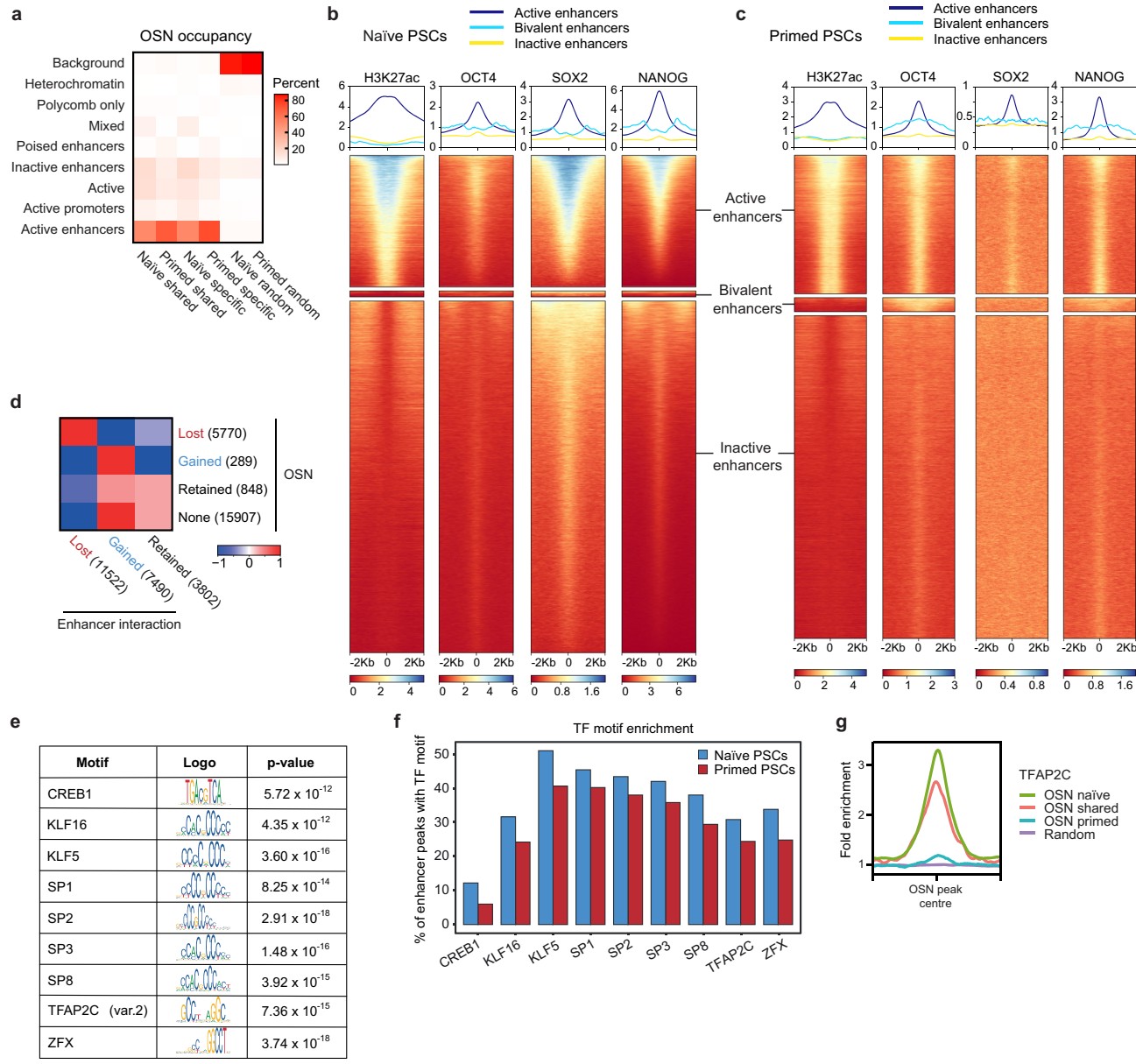

**Fig. 6 Widespread reorganisation of OCT4, SOX2 and NANOG binding at enhancers occurs between human pluripotent states. a** Heatmap shows the percentage of OSN sites that fall within each of the ChromHMM-defined chromatin states. Columns 1 and 2 indicate OSN sites that are common to both naive and primed PSCs; columns 3 and 4 are OSN sites that are specific to either naive or primed PSCs; columns 5 and 6 represent random regions that do not contain OSN sites. **b, c** ChIP-seq data for H3K27ac, OCT4, SOX2 and NANOG in **b** naive and **c** primed PSCs. Metaplots (upper) and heatmaps (lower) show normalised ChIP-seq read counts within a 4 kb peak-centred window. Regions were subsetted into active enhancers (naive, $n = 43,966$; primed, $n = 54,633$), bivalent enhancers (naive, $n = 1,998$; primed, $n = 5,502$) and inactive enhancers (naive, $n = 117,609$; primed, $n = 127,146$) based on ChromHMM-defined chromatin states, and ranked by H3K27ac signal. **d** Heatmap shows the log2 odds ratio for the associated changes in OSN occupancy and promoter–enhancer interactions in primed compared to naive PSCs. **e** Table shows the highest-ranking (by adjusted *P*-value, one-tailed Fisher's exact test) transcription factor motifs that are enriched at OSN sites in naive compared to primed PSCs. Four motifs associated with transcription factors that are not expressed in naive PSCs (log2 RPKM < 0) were removed from the list: KLF1, NR2F1, ZNF354C and VDR. **f** Bar chart shows the percentage of OSN-bound enhancers that contain each of the identified transcription factor motifs in naive and primed PSCs. **g** Enrichment of TFAP2C ChIP-seq signal across OSN peaks and the 2 kb up- and downstream regions.

these differences were associated with the widespread reorganisation of transcription factor binding (Fig. 7). The shared factors OCT4, SOX2 and NANOG bound predominantly to active enhancers in both pluripotent states. There was, however, a remarkable lack of overlap in OSN-occupied enhancers between naive and primed PSCs. The remodelling of OSN binding is likely to occur at a late stage of primed to naive PSC resetting because PSCs that are only partially reset towards the naive state show far

fewer differences in OCT4 and NANOG occupancy compared to primed PSCs[73]. Our results suggest that there is a substantial reorganisation of gene regulatory elements between human pluripotent states. In mouse PSCs, enhancer activation, SE interactions and OSN occupancy are also dynamic between pluripotent states[69,74–76] and these processes are modulated by the presence of other state-specific transcription factors such as ESRRB in naive cells and OTX2 and GRHL2 in primed

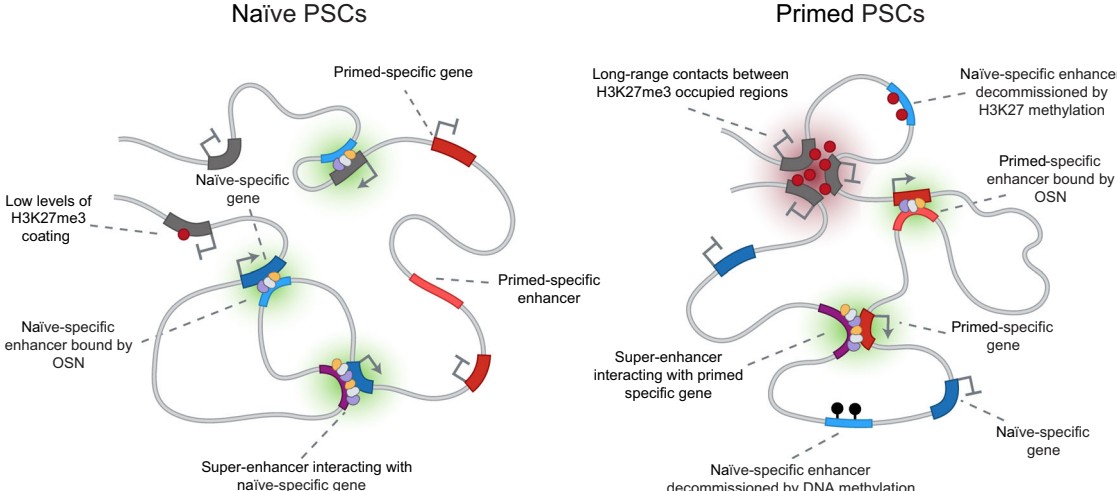

**Fig. 7 Schematic model that summarises the widespread reorganisation of gene regulatory interactions between naive and primed PSCs.** A substantial rewiring of promoter interactions occurs between naive and primed PSCs. Very few active enhancers are shared between naive (left) and primed (right) PSCs, and this divergence corresponds to important differences in the activity of key cell identity genes. Active enhancers in naive PSCs (light blue) are decommissioned predominantly through the gain of DNA methylation (black circles) during the transition into a primed state, and a minority acquire H3K27me3 (red circles) instead. The differences in enhancer activity are associated with the widespread reorganisation of OCT4, SOX2 and NANOG pluripotency factor binding, suggesting the presence of distinct, transcription factor-defined gene regulatory networks between PSC states. Many promoters were linked by long-range H3K27me3-associated interactions that formed de novo during the transition from the naive to the primed state, creating a strong spatial network of over 600 developmental genes.

cells[74,77–79]. *ESRRB* is not expressed in naive human PSCs nor in human epiblast cells in vivo[80–82], suggesting that in some instances the control of enhancer activity differs between mouse and human pluripotent states and that alternative factors are involved in human cells. One such factor is TFAP2C, which facilitates the opening of a subset of naive-specific enhancers in human PSCs but only has a modest role in mouse PSCs[55]. The emerging picture is that substantial rewiring of enhancer activity and interactions occurs during pluripotent state transitions and this is driven by a combination of common and species-specific transcription factors. It is important to note that not all regions marked by active enhancer modifications and transcription factor binding can drive gene expression in an ectopic reporter assay[73]. Potential differences in gene regulatory interactions between individual cells within a population will also be masked using current, bulk approaches. Further complexity is that the cooperative behaviour of multiple transcription factors seems to be context-dependent in terms of which enhancers are targeted, and this complexity will need to be unravelled through precise functional analyses. Our results will transform the interpretation of these experiments by comprehensively identifying the target gene promoters for enhancers and their dynamic alteration in human pluripotent states.

Network approaches have been applied successfully to global Hi-C data by using network modularity to identify communities of interacting loci that represent high-level chromatin features such as TADs and sub-TADs[40–42], and principles of chromatin assortativity have also been used to represent capture-based Hi-C data[43]. The Canvas network approach that we developed here represents patterns of interactions (edges) between genomic regions (nodes), with added edge weights and node values that can convey user-defined information such as cell-type specificity, the chromatin or transcriptional state of each region, or the interaction strength. We chose to visualise the combined network using a force-directed layout[47]. Other visualisation algorithms that we tested were less suitable because they generated network graphs that lacked informative structure or were incompatible with large-scale datasets. The final, constructed networks revealed

prominent features of genome organisation at multiple scales such as TADs and chromatin loops. Canvas, therefore, provides a powerful and intuitive data exploration method to understand the relationships behind the emergent connectivity patterns. Importantly, Canvas can also view the interaction map for the whole genome in one plot whilst retaining relevant topological features, and can be overlaid with other, user-defined datasets. Our network-scale approach could be applied to other interaction datasets and we provide our scripts and documentation to facilitate this. Future iterations of the network graph could take into account other features of spatial organisation such as chromosome territories and positioning, and lamin-associated domains, to instruct the layout of nodes within the network (Supplementary Fig. 10). The network could also be customised by including edge weights, for instance, based on chromatin interaction scores or distances (Supplementary Fig. 10). Another area of future development could be to adapt the existing network to make predictions about how the set of interactions is formed or how the interactions might change upon perturbation or cell state changes. Such theory-based models typically require a higher level of abstraction away from the experimental findings but are able to make network-dependent predictions about the mechanisms and dynamics of the interaction model.

Our study provides an invaluable resource to study the complex and multilayered interplay between transcription factors, chromatin state and 3D genome organisation in controlling cell state. The experimental and computational approach set out here could be applied to other high-resolution interaction datasets to uncover topological changes that underpin cell state transitions during development and disease.

## Methods
Reagent and resource details are provided in Supplementary Dataset 4.

**Cell lines**. WA09/H9 NK2 naive and primed PSCs were kindly provided by Dr. Austin Smith[22] with permission from WiCell and the UK Stem Cell Bank Steering Committee. All PSCs were cultured in 5% $O_2$, 5% $CO_2$ at 37 °C. Because some naive PSCs have been reported previously to acquire tetraploidy over time, we used Hi-C sequencing reads to examine copy number variation in the naive and

primed samples used in this current study. The results in Supplementary Fig. 11 show that the NK2 naive PSCs have a normal copy number across nearly all chromosomes. The main alteration is on chromosome 19, and this alteration is present in both naive and primed PSCs. We also examined the chromosome copy number of cells by DNA FISH, which showed that <10% of naive PSCs were tetraploid. These data indicate that the PSCs used in this study have good genome stability.

**Cell culture.** Naive PSCs were maintained in t2iL + PKCi media as previously described[22] in a 1:1 mixture of DMEM/F12 and Neurobasal, 0.5× N2 supplement, 0.5× B27 supplement, 1× nonessential amino acids, 2mM L-Glutamine, 1× Penicillin/Streptomycin (all from ThermoFisher Scientific), 0.1 mM β-mercaptoethanol (Sigma-Aldrich), 1 µM PD0325901, 1 µM CHIR99021, 20 ng/ml human LIF (all from WT-MRC Cambridge Stem Cell Institute) and 2 µM Gö6983 (PKCi; Tocris) on Matrigel-coated plates (Corning). Primed PSCs were maintained on Vitronectin-coated plates (0.5 µg/cm²; ThermoFisher Scientific) in TeSR-E8 medium (StemCell Technologies).

**Hi-C.** Approximately 35 million naive or primed PSCs were fixed in 2% formaldehyde (Agar Scientific) for 10 min, after which the reaction was quenched with ice-cold glycine (Sigma; 0.125 M final concentration). Cells were collected by centrifugation (400×g for 10 min at 4 °C), and washed once with 50 ml PBS pH 7.4 (Gibco). After another centrifugation step (400×g for 10 min at 4 °C), the supernatant was completely removed and the cell pellets were immediately frozen in liquid nitrogen and stored at −80 °C. After thawing, the cell pellets were incubated in 50 mL ice-cold lysis buffer (10 mM Tris-HCl pH 8, 10 mM NaCl, 0.2% Igepal CA-630, cOmplete EDTA-free protease inhibitor cocktail (Roche) for 30 min on ice. After centrifugation to pellet the cell nuclei (650×g for 5 min at 4 °C), nuclei were washed once with 1.25×NEBuffer 2 (NEB). The nuclei were then resuspended in 1.25×NEBuffer 2, SDS (10% stock; Promega) was added (0.3% final concentration) and the nuclei were incubated at 37 °C for one hour with agitation on an Eppendorf Thermomixer (950 rpm). Triton X-100 (Sigma) was added to a final concentration of 1.7% and the nuclei were incubated at 37 °C for one hour with agitation on an Eppendorf Thermomixer (950 rpm). Restriction digest was performed overnight at 37 °C with agitation (950 rpm) with HindIII (NEB; 1500 units per 7 million cells). Using biotin-14-dATP (Life Technologies), dCTP, dGTP and dTTP (Life Technologies; all at a final concentration of 30 µM), the HindIII restriction sites were then filled in with Klenow (NEB) for 75 min at 37 °C, followed by ligation for 4 h at 16 °C (50 units T4 DNA ligase (Life Technologies) per 7 million cells starting material) in a total volume of 5.5 mL ligation buffer (50 mM Tris-HCl, 10 mM MgCl2, 1 mM ATP, 10 mM DTT, 100 µg/ml BSA) per 7 million cells starting material. After ligation, crosslinking was reversed by incubation with Proteinase K (Roche; 65 µl of 10 mg/ml per 7 million cells starting material) at 65 °C overnight. An additional Proteinase K incubation (65 µl of 10 mg/ml per 7 million cells starting material) at 65 °C for 2 h was followed by RNase A (Roche; 15 µl of 10 mg/ml per 7 million cells starting material) treatment and two sequential phenol/chloroform (Sigma) extractions. After DNA precipitation (sodium acetate 3 M pH 5.2 (1/10 volume) and ethanol (2.5 × volumes)) overnight at −20 °C, the DNA was spun down (centrifugation 3200 x g for 30 min at 4 °C). The pellets were resuspended in 400 µl TLE (10 mM Tris-HCl pH 8.0; 0.1 mM EDTA), and transferred to 1.5 ml Eppendorf tubes. After another phenol/chloroform (Sigma) extraction and DNA precipitation overnight at −20 °C, the pellets were washed three times with 70% ethanol, and the DNA concentration was determined using Quant-iT Pico-Green (Life Technologies). For quality control, candidate 3C interactions were assayed by PCR, and the efficiency of biotin incorporation was assayed by amplifying a 3C ligation product, followed by a digest with HindIII or NheI.

To remove biotin from non-ligated fragment ends, 40 µg of Hi-C library DNA were incubated with T4 DNA polymerase (NEB) for 4 h at 20 °C, followed by phenol/chloroform purification and DNA precipitation overnight at −20 °C. After one wash with 70% ethanol, sonication was carried out to generate DNA fragments with a size peak around 400 bp (Covaris E220 settings: duty factor: 10%; peak incident power: 140 W; cycles per burst: 200; time: 55 s). After end repair (T4 DNA polymerase, T4 DNA polynucleotide kinase, Klenow (all NEB) in the presence of dNTPs in ligation buffer (NEB)) for 30 min at room temperature, the DNA was purified (Qiagen PCR purification kit). dATP was added with Klenow exo- (NEB) for 30 min at 37 °C, after which the enzyme was heat-inactivated (20 min at 65 °C). A double size selection using AMPure XP beads (Beckman Coulter) was performed: first, the ratio of AMPure XP beads solution volume to DNA sample volume was adjusted to 0.6:1. After incubation for 15 min at room temperature, the sample was transferred to a magnetic separator (DynaMag-2 magnet; Life Technologies), and the supernatant was transferred to a new Eppendorf tube, while the beads were discarded. The ratio of AMPure XP beads solution volume to DNA sample volume was then adjusted to 0.9:1 final. After incubation for 15 min at room temperature, the sample was transferred to a magnet (DynaMag-2 magnet; Life Technologies). Following two washes with 70% ethanol, the DNA was eluted in 100 µl of TLE (10 mM Tris-HCl pH 8.0; 0.1 mM EDTA). Biotinylated ligation products were isolated using MyOne Streptavidin C1 Dynabeads (Life Technologies) on a DynaMag-2 magnet (Life Technologies) in binding buffer (5 mM Tris pH 8, 0.5 mM EDTA, 1 M NaCl) for 30 min at room temperature. After two washes in binding buffer and one wash in ligation buffer (NEB), pre-annealed PE adapters 1 and 2 (Illumina; Supplementary Dataset 4) were ligated onto Hi-C ligation products

bound to streptavidin beads for 2 h at room temperature (T4 DNA ligase NEB, in ligation buffer, slowly rotating). After washing twice with wash buffer (5 mM Tris, 0.5 mM EDTA, 1 M NaCl, 0.05% Tween- 20) and then once with binding buffer, the DNA-bound beads were resuspended in a final volume of 90 µl NEBuffer 2. Bead-bound Hi-C DNA was amplified with 7 PCR amplification cycles using PE PCR 1.0 and PE PCR 2.0 primers (Illumina; Supplementary Dataset 4). After PCR amplification, the Hi-C libraries were purified with AMPure XP beads (Beckman Coulter). The concentration of the Hi-C libraries was determined by Bioanalyzer profiles (Agilent Technologies), and the Hi-C libraries were paired-end sequenced (HiSeq 2500, Illumina).

**Promoter Capture Hi-C.** 500 ng of Hi-C library DNA was resuspended in 3.6 µl water, and hybridisation blockers (Agilent Technologies; hybridisation blockers 1 and 2, and custom hybridisation blocker) were added to the Hi-C DNA. Hybridisation buffers and the custom-made RNA capture bait system (Agilent Technologies; designed as previously described[38]: 37,608 individual biotinylated RNAs targeting the ends of 22,076 promoter-containing human HindIII restriction fragments) were prepared according to the manufacturer's instructions (SureSelect Target Enrichment, Agilent Technologies). The Hi-C library DNA was denatured for 5 min at 95 °C, and then incubated with hybridisation buffer and the RNA capture bait system at 65 °C for 24 h (all incubation steps in an MJ Research PTC-200 PCR machine). After hybridisation, 60 µl of MyOne Streptavidin T1 Dynabeads (Life Technologies) were washed three times with 200 µl binding buffer (SureSelect Target Enrichment, Agilent Technologies), before incubation with the Hi-C DNA/RNA capture bait mixture with 200 µl binding buffer for 30 min at room temperature, slowly rotating. Hi-C DNA bound to capture RNA was isolated using a DynaMag-2 magnet (Life Technologies). Washes (15 min in 500 µl wash buffer I at room temperature, followed by three 10 min incubations in 500 µl wash buffer II at 65 °C) were performed according to the SureSelect Target enrichment protocol (Agilent Technologies). After the final wash, the beads were resuspended in 300 µl NEBuffer 2, isolated on a DynaMag-2 magnet, and then resuspended in a final volume of 30 µl NEBuffer 2. After a post-capture PCR (four amplification cycles using Illumina PE PCR 1.0 and PE PCR 2.0 primers; 13 to 15 individual PCR reactions), the Promoter CHi-C libraries were purified with AMPure XP beads (Beckman Coulter). The concentration of the Promoter CHi-C libraries was determined by Bioanalyzer profiles (Agilent Technologies), and the Promoter CHi-C libraries were paired-end sequenced (HiSeq 2500, Illumina).

**Hi-C analysis.** HiCUP[83] was used to map and filter di-tags to the human genome build GRCh38. We compared the global organisation by plotting the log10 frequency of cis-chromosomal contacts in the raw data at various genomic distances on the log10 scale.

TADs were identified based on directionality indices of Hi-C interactions[8], using binned Hi-C data at 25 kb resolution and sliding bins every 5 kb using HOMER v4.7 with minDelta=2 and other parameters kept at their default values. This resulted in 3,124 TADs for naive PSCs and 2,917 TADs for primed PSCs.

Hi-C peaks in Fig. 2 were identified using HiCCUPS v1.8.8[12] employing Knight–Ruiz balancing by Juicer tools as the matrix balancing algorithm with the following parameters: `--ignore_sparsity -k KR -f 0.1 -r 250000 -d 750000 -i 8 -p 4`. A combined list of genome-wide peaks identified in both naive and primed PSCs were used for aggregate peak analysis with the following parameters: `-r 250000 -n 30 -w 10` using Juicer tools v1.8.9[84]. Additional aggregate pileup plots were constructed using coolpup.py[85].

**Promoter capture Hi-C analysis.** PCHi-C data were mapped and filtered using HiCUP[83] with the GCRh38 human genome build. CHiCAGO[45] was used to define significant promoter interactions at the level of individual HindIII fragments. CHiCAGO uses a convolution background model for the background level of interactions for pooled baited and other regions, and a weighted distance-dependent multiple testing correction. The background model is a convolution of negative binomial (a genomic distance-dependent term modelling Brownian collisions) and Poisson distributions (a distance-independent term modelling technical noise). The weighted distance-dependent multiple testing correction generated using the reproducibility of interactions between the replicates makes CHiCAGO robust in the presence of undersampling, in regions with low counts, providing a rigorous statistical framework for identifying interactions. Two biological replicates for each cell type were normalised and combined as part of the CHiCAGO pipeline. CHiCAGO interaction scores correspond to –log-transformed, weighted P-values for each fragment read pair. A CHiCAGO interaction score of 5 or above was considered significant based on previous empirical observations[19,45]. The network graph was constructed using promoter interactions with scores of 5 or above. Figures 4 and 6 focus predominantly on investigating promoter–enhancer communication in naive and primed PSCs, with a view to identifying shared and state-specific gene regulatory interactions. An initial concern that we had when working on these comparisons was that our standard analysis would be too sensitive in calling differences in interactions between cell types if we applied a strict cutoff of a CHiCAGO score of >5. We therefore adopted a more cautious approach of allowing interaction to be called if it had a CHiCAGO score of >3 in one cell type and a score >5 in the other cell type. That way we could

be more confident that the identified differences in interactions between cell types were likely to be robust and not due to a minor sub-threshold score.

**Network construction and analysis.** PCHi-C interactions with CHiCAGO scores of 5 or more were used to generate the network graph using custom scripts in R and Python. Promoter regions were defined as 1 kb upstream of the TSS based on the Ensembl v85 gene model annotation and assigned to HindIII fragments. Interaction networks were constructed using the igraph v1.2.1 R package[86] where each HindIII fragment represents a node and each significant interaction represents an edge. The edges of both naive and primed PSCs were combined into a single network with a common layout. The shared anchor points allow the examination of interactivity of shared genomic regions. For Supplementary Fig. 3, community detection was performed using the multi-level optimisation algorithm[48] implemented within igraph. Sub-networks with a modularity score of 0.7 or above were split into individual communities. The coordinates of HindIII fragments with the assigned communities were compared to TAD coordinates obtained from Hi-C analysis using the GenomicRanges R package v1.30.3[87,88], while permutation tests were performed with the RegioneR R package[89]. Network visualisation was performed with Gephi v0.9.2[90] using the ForceAtlas2 layout algorithm[47]. For Fig. 2b, the multi-dimensional scaling (MDS) layout within Gephi, developed by Wouter Spekkink (http://www.wouterspekkink.org), was used to obtain linear genomic distance representative layouts of individual sub-networks.

For Supplementary Fig. 6c, interaction distances >1.5 times the interquartile range were classified as outliers and not plotted.

**3D DNA Fluorescent in situ hybridisation (FISH).** Briefly, 50–200 μl of cells at a concentration of 10 million cells/ml were placed onto poly-L-lysine coated slides (Sigma) and left for 5 min to allow attachment. The slides were placed into 4% formaldehyde/PBS (PFA; VWR) for 10 min and quenched in glycine for at least 10 min at 22 °C. After permeabilisation using Saponin (Sigma), the slides were stored in 50% glycerol at −20 °C for at least a week before proceeding to three rounds of freeze-thawing in liquid nitrogen. After PBS (Gibco) washes, the slides were placed in 0.1 N HCl, rinsed and then subjected to another round of permeabilisation using Saponin and Triton X-100 (Sigma). Probes precipitated with human Cot-I (ThermoFisher Scientific) and salmon sperm DNA (Sigma) was added to the cells, which subsequently underwent denaturation at 78 °C for exactly 2 min, followed by a 16 h incubation at 37 °C to allow hybridisation. After SSC washes, cells were counterstained with DAPI followed by an additional fixation with 3.7% formaldehyde, which results in a cleaner signal and longer storage life. Finally, a coverslip was mounted along with a drop of Vectashield (Vector Laboratories) or ProLong Diamond antifade mounting media (ThermoFisher Scientific). Imaging was performed on the Metafer/MetaCyte slide scanning system.

**Direct labelling of fluorescent FISH probes.** BAC DNA was extracted using the NucleoBond BAC 100 kit (Macherey-Nagel), following the manufacturer's protocol. Probes used: DLX1/2 (M22), HOXD10/11 (F14) and control locus (RP11-297L12). Nick translation labelling of probes was performed as described[91] with some modifications. Briefly, mixed on ice 5 μl 10× NTB (0.5 M Tris-HCl pH 7.5, 50 mM MgCl₂, 0.5 mg/ml nuclease-free BSA fraction V), 0.1 M DTT, 4 μl d(GAC) TP mix 0.5 mM, 1 μl dTTP 0.5 mM, 6 μl aminoallyl-dUTP 0.5 mM (ThermoFisher Scientific), 1 μl DNA Polymerase I 10 U/μl (NEB), 1 μl DNase I dilution (1:25) (Roche), H₂O to a final volume of 50 μl. The mix was incubated for exactly 2 h at 16 °C and then for 5 min at 75 °C to inactivate DNase I. DNA digestion was visualised using a 2% agarose gel. Each probe was nick-translated in a separate reaction (1 μg each). Optimally digested DNA was pooled (4 μg), ethanol precipitated, and the pellet was re-suspended in four times 1.25 μl H₂O followed by quantification using the NanoDrop spectrophotometer. Single-use dried pellets of amine-reactive dye were re-suspended in 2 μl anhydrous DMSO (Sigma). For a 10 μl reaction, 2 μg of DNA in a 5 μl volume was heated to 95 °C for 5 min and placed on ice. 3 μl of NaB buffer (0.2 M Sodium bicarbonate pH 8.3) was added to snap cooled amine-modified DNA followed by 2 μl of re-suspended dye. After a 1-h incubation, the QIAquick PCR purification kit was used to purify the probes. The probe fluorescence intensity was analysed on the NanoDrop 1000.

**FISH analysis.** Automated spot identification and their positional coordinates in relation to the nuclear volume were obtained using the Metafer software. Cells without two spots for each FISH probe were rejected. DAPI staining was used to determine nuclear volume. Cartesian coordinates of spots were exported from the Metafer software and distances between all signal pairs were calculated using a custom script. The shortest distance in each cell was used in the final analysis.

**Chromatin immunoprecipitation.** All buffers were pre-chilled to 4 °C with cOmplete EDTA-free protease inhibitor (Roche) freshly added. 15 million cells per ChIP were treated with Accutase (ThermoFisher Scientific) and collected in a 50 ml conical tube, followed by a 300×g 5 min spin at 4 °C. The pellets were resuspended in PBS. Cells were cross-linked with 2 μM DSG (Sigma) for 45 min at 22 °C and then with 1% methanol-free formaldehyde (Agar Scientific) at a cell density of 10⁸ cells in 45 ml media for 12.5 min at 22 °C. Fixation was stopped with the addition of glycine at a final concentration of 125 mM and incubation for 5 min at 22 °C.

After two PBS washes, cells were resuspended in Wash buffer 1 (10 mM Hepes pH 7.5; 10 mM EDTA; 0.5 mM EGTA; 0.75% Triton X-100) and incubated for 10 min at 4 °C. After spinning at 3200×g for 5 min at 4 °C, nuclei were resuspended in 10 ml Wash buffer 2 (10 mM Hepes pH 7.5; 200 mM NaCl; 1 mM EDTA; 0.5 mM EGTA) and incubated for 10 min at 4 °C. Another 3200×g 5 min spin at 4 °C was performed followed by resuspension in 1 ml of freshly made Lysis/sonication buffer (150 mM NaCl; 25 mM Tris pH 7.5; 5 mM EDTA; 0.1% Triton X-100; 1% SDS; freshly dissolved 0.5% Sodium deoxycholate) per 12 million cells. Lysis was performed on ice for 30 min, followed by sonication for 15 s on, 30 s off (Microson ultrasonic cell disruptor XL Misonix; output setting 4; 10–11 W) for 20 cycles to obtain fragments with a size of 200–500 bp. Fragmented chromatin was spun down at 10,000×g for 15 min at 4 °C, and the supernatant was transferred to a new tube and diluted 1:10 with ChIP dilution buffer (150 mM NaCl; 25 mM Tris pH 7.5; 5 mM EDTA; 1% Triton X-100; 0.1% SDS; 0.5% Sodium deoxycholate). 500 μl was taken for the input and the remaining diluted supernatant was incubated with primary antibody overnight at 4 °C. Antibodies were against NANOG (R&D; AF1997; 5 μg per ChIP), SOX2 (R&D; AF2018; 5 μg per ChIP) and IgG (Jackson ImmunoResearch; 315-005-003; 5 μg per ChIP). Magnetic protein A (120 μl per IP) or protein G (180 μl per IP) Dynabeads (both from Invitrogen) were washed with Wash buffer A (50 mM Tris pH 8; 150 mM NaCl; 0.1% SDS; 0.5% Sodium deoxycholate; 1% NP40; 1 mM EDTA) and blocked for 1 h at 4 °C with yeast tRNA (Invitrogen) and BSA (NEB). The pre-blocked beads were added to the antibody-bound chromatin and incubated for 7–8 h at 4 °C. Subsequently, the magnetic beads with the bound antibody–chromatin-complex were rinsed once with Wash buffer A, washed twice with Wash buffer A, washed once with Wash buffer B (50 mM Tris pH 8.0; 500 mM NaCl; 0.1% SDS; 0.5% Sodium deoxycholate; 1% NP40; 1 mM EDTA), washed once with Wash buffer C (50 mM Tris pH 8; 250 mM LiCl; 0.5% Sodium deoxycholate; 1% Igepal CA-630; 1 mM EDTA) and rinsed with 1× TE buffer (10 mM Tris pH 8; 1 mM EDTA). Chromatin was eluted off the beads with 450 μl of Elution buffer (1% SDS; 0.1 M NaHCO₃). Additionally, 11 μl Proteinase K (20 mg/ml) and 5 μl RNase A (10 mg/ml) were added (including to the input) and incubated at 37 °C for 2 h, followed by overnight incubation at 65 °C to reverse the crosslink. DNA was purified using AMPure XP beads (Beckman Coulter) and eluted in 40 μl water. DNA was quantified using the Qubit fluorometer dsDNA HS assay kit (ThermoFisher Scientific). Libraries were prepared using the NEBNext Ultra II DNA library prep kit for Illumina (NEB) using the manufacturer's protocol.

**CUT&RUN.** CUT&RUN was performed on 50,000 cells per biological replicate, following the protocol published by the Henikoff lab[92]. Cells were harvested using Accutase and washed twice in 20 mM HEPES pH 7.5; 150 mM NaCl; 5 mM Spermidine. Cells were bound to Concanavalin A beads (Bangs Laboratories; 10 μl per sample, beads were washed twice beforehand with binding buffer (20 mM HEPES-KOH pH 7.9; 10 mM KCl; 1 mM CaCl₂; 1 mM MnCl₂)). Cells were simultaneously permeabilised and incubated with an antibody raised against H3K4me1 (Abcam; ab8895; 5 μg per ChIP) for 10 min by the addition of 50 μl of Digitonin buffer (20 mM HEPES pH 7.5; 150 mM NaCl; 5 mM Spermidine; 0.1% Digitonin) containing the antibody at a ratio of 1 in 50. After washing with Digitonin buffer, the sample was incubated with pA-MNase in Digitonin buffer (final concentration 700 ng/ml) for 10 min and then washed twice with Digitonin buffer. Targeted cleavage was induced in 150 μl Digitonin buffer containing 2 mM CaCl₂ for 30 min on ice. The reaction was stopped by the addition of 2× Stopp buffer (0.34 M NaCl; 20 mM EDTA; 4 mM EGTA; 0.02% Digitonin; 0.05 mg/ml RNaseA; 0.05 mg/ml glycogen). Cleaved fragments were released by 10 min incubation at 37 °C. The DNA was Proteinase K digested and extracted by Phenol/Chloroform extraction before being Ethanol precipitated and subjected to library preparation using NEBNext Ultra II DNA Library Prep Kit for Illumina (NEB) according to manufacturer's guidelines.

**Genome browser tracks.** Normalised bigwig files for genome browser visualisation were produced using Deeptools[93]. For ChIP-seq and CUT&RUN samples, the BAM files were normalised with the reads per genomic content (RPGC) method, ignoring chrY and chrMT. A 10 bp bin size with a 200 bp read extension was chosen with a bigwig file output. The experimental inputs were subtracted from each sample. For RNA-seq tracks, the BAM files were normalised using the DESeq2 scaling factor (Naive - 0.34576529; Primed - 1.873937595), with a default bin size of 50 bp. Genome browser tracks were visualised using the WashU Epigenome Browser v48.2.0+[94–96].

**Quantification and statistical analysis**

*ChIP-sequencing and CUT&RUN analysis.* Reads were trimmed using Trim Galore and mapped to the human genome GRCh38 using Bowtie2[97]. All analyses were performed using SeqMonk and R. ChIP-seq peaks were called using MACS2 [98] with parameters $q < 10^{-9}$ for all histone modification samples except for H3K4me1 for which the cutoff used was $q < 10^{-7}$. For quantitation, read lengths were extended to 300 bp and regions of coverage outliers were excluded. Quantitations (log2 RPM) for all analysed regions and histone marks are provided in the OSF (ChIP_Quantiation_revision.xlsx). OCT4, NANOG and SOX2 peaks were called using a SeqMonk implementation of MACS[98] with parameters $p < 10^{-5}$, sonicated fragment size = 300. Peaks were filtered by signal intensity, retaining only peaks that overlap with at least one 500 bp window in which log2 RPM > 0. Regions of

OCT4, NANOG and SOX2 peaks were combined and merged if closer than 100 bp. The resulting list of regions was filtered for those that overlap with MACS peaks for all three factors and called OSN peaks. Control regions are 10,000 randomly selected 1.2 kb windows (approximate average peak size). To assign one ChromHMM state per peak, the location of the peak centre was used. TFAP2C peaks were called using a SeqMonk implementation of MACS[98] with parameters $p < 10^{-5}$, sonicated fragment size = 300 for individual replicates, and the overlap between replicates of the resulting regions were used for quantitation.

In Fig. 4a–h, enhancers and super-enhancers were annotated using ROSE[50,51] with H3K27ac peaks called using MACS2 with parameters $q < 10^{-9}$. A stitching distance of 1.5 kb was chosen based on the bimodal distribution of the distance to the nearest peak.

*Chromatin state annotation—ChromHMM.* Chromatin state analysis was performed using ChromHMM[46]. Trim Galore quality trimmed and Bowtie2 aligned (GRCh38) BAM files were binarized using the `BinarizeBam` command with default 200 bp bin settings. Naive and primed PSCs were stacked to provide a single-genome annotation with the inclusion of ChIP-seq input samples as an additional feature. Model learning was performed on a range of states with 16 being selected as the final number. These categories were reduced to seven states that were more biologically relevant (active—H3K27ac, H3K4me3, H3K4me1; Polycomb-repressed—H3K27me3; bivalent—H3K27me3, H3K4me3, H3K4me1; heterochromatin repressed—H3K9me3; unclassified—H3K27me3, H3K4me3, H3K4me1, H3K9me3; background—low emission probability levels in all samples).

To assign a state to each *Hin*dIII fragment, the overlap of the seven states with each *Hin*dIII restriction fragment was determined and reduced to a single final chromatin state based on the following rules: any single state superseded the background state; the bivalent state superseded the Polycomb-repressed state, and a mixture of multiple states was labelled as mixed. Genomic features relating to ChromHMM states were used in Figs. 5 and 6 and their definitions are summarised in Supplementary Dataset 3.

*RNA-seq analysis.* RNA-sequencing reads from ref. 22 were trimmed using Trim Galore 0.3.8 using default parameters to remove the standard Illumina adaptor sequence. Reads were mapped to the human GRCh38 genome assembly using HISAT 2.0.5[99] guided by the gene models from the Ensembl v85 release. Samtools[100] was used to convert to BAM files that were imported to Seqmonk. Raw read counts per transcript were calculated using the RNA-sequencing quantitation pipeline on the Ensembl v85 gene set using directional counts. Differentially expressed genes were determined using DESeq2[101]. Log2(FPKM) normalised values were generated with Seqmonk.

*DNA methylation analysis.* Whole-genome bisulfite sequencing data from ref. 22 was trimmed using Trim Galore 0.3.8, aligned with Bismark v0.18.2[102] and analysed in SeqMonk. Methylation is given as a percentage of methylated cytosine calls overall observations for each enhancer or background region.

*CNV analysis.* CNV analysis was performed using the HiNT-CNV tool[103]. Cooler[104] unnormalized Hi-C data at 100 kb resolution were used as input. The 50mer UCSC track was used for Hi-C bias removal.

*Gene ontology analysis.* Gene ontology analysis of protein-coding genes within the H3K27me3-associated interaction network in primed PSCs was performed using Enrichr[105,106].

*Motif enrichment.* To find differentially enriched motifs between naive and primed-specific OSN binding sites, sequences 250 bp up- and downstream of the peak centre were repeat masked (RepeatMasker, A.F.A. Smit, R. Hubley & P. Green, version open-4.0.9, default parameters). 500 OSN peaks were randomly selected per category and analysed using AME (version 5.0.5[107]) with the following parameters (naive over primed): ame --verbose 1 --oc. --scoring avg --method fisher --hit-lo-fraction 0.25 --evalue-report-threshold 10.0 --control 500_random_primed_specific_OSN.txt.fa 500_random_naive_specific_OSN.txt.fa db/JASPAR/JASPAR2018_CORE_vertebrates_non-redundant.meme. For enrichment 'primed over naive' the two input files were swapped. Resulting lists of enriched motifs were filtered for expression of the respective binding factor in at least one of the two PSC states (log2 RPKM > 0). A FIMO search[108] using the sequences of all selected OSN regions as a background model was then performed to determine whether an individual OSN peak contained the motif or not.

*Reporting summary.* Further information on research design is available in the Nature Research Reporting Summary linked to this article.

## Data availability
Sequence data that support the findings of this study have been deposited in Gene Expression Omnibus with the primary accession code GSE133126. Processed data have been made available through the Open Science Framework (https://osf.io/jp29m)[109]. All other relevant data supporting the key findings of this study are available within the article and its Supplementary Information files or from the corresponding author upon reasonable request. A reporting summary for this Article is available as a Supplementary Information file.

## Code availability
Custom scripts are available from GitHub (https://github.com/peterch405/pchic_network)[110].

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

## Acknowledgements
We thank Steven Wingett, Felix Krueger, Anne Segonds-Pichon and Simon Andrews from Babraham Bioinformatics for sequencing QC, mapping, bioinformatic analysis and statistical advice; Simon Walker from the Babraham Institute Imaging Facility for help with DNA FISH experiments; and Kristina Tabbada and Clare Murnane of the Babraham Institute Next Generation Sequencing Facility. We are also grateful to Mikhail Spivakov and Jonathan Cairns for advice about CHiCAGO and Peter Fraser for support and helpful discussions. Work in our laboratories is supported by grants from the BBSRC (BBS/E/B/000C0421, BBS/E/B/000C0422, BB/J004480/1, BBS/E/B/000C0404, BBS/E/B/000C0405, BBS/E/B/000C0427, BBS/E/B/000C0428, Core Capability Grant). P.C. was supported by a BBSRC iCASE PhD studentship (1520397). A.J.C. was supported by an MRC DTG Studentship (MR/J003808/1). S.S. was supported by a UKRI MRC Rutherford Fund Fellowship (MR/T016787/1) and a Career Progression Fellowship from the Babraham Institute. C.V. was supported by an ERC AdG to Peter Fraser (DEVOCHROMO).

## Author contributions
Conceptualisation, P.C., A.J.C., S.S., A.E.C. and P.J.R.-G; methodology, P.C. and A.J.C; software, P.C; validation, P.C., A.J.C., C.K., C.V., S.S., A.E.C. and P.J.R.-G; formal analysis, P.C., A.J.C., C.K. and C.V; investigation, P.C., A.J.C., C.I.S. and S.S; resources, P.C., A.J.C., S.S., A.E.C. and P.J.R.-G; data curation, P.C., A.J.C., C.K. and C.V; writing—original draft, P.C., A.J.C. C.K., C.V. and P.J.R-G; writing—review and editing, P.C., A.J.C. C.K., C.V., C.I.S., S.S., A.E.C. and P.J.R.-G; visualisation, P.C., A.J.C. C.K. and C.V; supervision, S.S., A.E.C. and P.J.R.-G; project administration, P.C., A.J.C., S.S., A.E.C. and P.J.R.-G; funding acquisition, S.S., A.E.C. and P.J.R.-G.

## Competing interests
S.S. is a co-founder of Enhanc3D Genomics Ltd. The remaining authors declare no competing interests.
