## [Peer Review File · Nature Communications]

REVIEWER COMMENTS

Reviewer #1 (Remarks to the Author):

In this manuscript “Widespread reorganisation of pluripotent factor binding and gene regulatory interactions between human pluripotent states” by Chovanec et al. the authors report on a new way to visualize and explore functional chromatin interactions between two distinct pluripotency stages in human cells, naive and primed. This is a very interesting manuscript, of interest beyond the stem cell field. This developmental stage is unique in exploring how cellular identities are established. What I found in particularly exciting is that the data suggests that the two pluripotent states have large clusters of interacting genes, which might have functional relevance, and also because forward differentiation from naive to primed is forming de novo long range interactions, specifically in bivalent regions.

I have a couple of significant issues.

1. I haven't seen any molecular experimental validation of the findings. Not being an expert in Hi-C methodologies, is there so much confidence in these pipelines that it doesn't warrant independent experimental validation of strong interactions that change between naive and primed? The data is very interesting and the visualisation is remarkable, but it would be highly desirable to have some molecular data to corroborate the large scale dataset findings, so far this is at best a potentially excellent hypothesis generation technology. For example, can one not visualize the H1 clusters, claimed to be specific for naive cells and protocadherin, claimed to be specific for primed cells, using DNA FISH? A further suggestion, if feasible: it would be interesting to see the temporal resolution of these interactions and the kinetics of this process (ie. to establish whether this is a quick change or a gradual change).

2. Regarding DNA methylation and the title of the subsection “Naive-specific active enhancers are decommissioned predominantly 1 by DNA methylation” The data shows a correlation indeed, but the title would suggest that a DNMT3a/b KO or KD cell line was used to demonstrate that active enhancers are decommissioned by DNA methylation. This is a very interesting hypothesis, which would imply that in the absence of DNA methyltransferases (KO, KD or chemical inhibition) the majority of enhancers would not be decommissioned. If this is not feasible then only correlation can be claimed, which is also a significant step. The wording will then need to be adjusted.

Further questions:

3. Regarding PCHI-C, are two biological replicates sufficient? There is a very significant correlation in Fig S1A, right panel, between the two pluripotent stages. What is the statistical uncertainty that any given point is outside the difference in replicates? One possible way to express this would be, for figure 1A(left) take the difference between the read counts of the replicates on the y axis, and the average of the read counts on the x axis. The standard error is then obvious from the y axis, and any variation in standard error versus average read count can also be calculated. Do the same for 1A(middle). Now for 1A(right) you can estimate, for every point, the z-score of the difference from primed to naive. Other methods may be even more appropriate, but the essential point is that you should be able to provide a much more robust argument for the difference being claimed. If a method is embedded in the pipelines, that needs to be clearly mentioned in the main text. Also notice that H3K4me1_naive ChIP-seq is done in only one replicate. Can the authors comment on how they arrived to using one or two replicates per sample?

4. Related to this, in Canvas, in the network graph, what accounts for a significant interaction “each edge represents a significant interaction between nodes” in Line 23? Also, I guess I don't get a sense of the population heterogeneity. Is significance a measure of increased interaction within a population? Can heterogeneity be inferred from this data in a specific locus? Stem cells are known to be heterogeneous, particularly primed, how can one distinguish heterogeneity from a non-significant interaction?

5. NK2 cells are known to acquire tetraploidy over time (a significant fraction of the population), which I assume will impact on chromatin interactions. Have the authors mitigate against this potential problem, can they comment?

Minor points:

1. Suppl Fig 2B Differentially expressed is clear but not clear in which direction

2. "This implies that the H3K4me1/3-marked sites are protected from H3K27me3 in naive PSCs, but that H3K27me3 spreads throughout the region in primed PSCs. Genes within these regions were associated with developmental processes, with examples including DLX, GATA and HOX factors." Can one see profiles of these genes/regions, in primed and naive, to see how "commonly with H3K4me1/3-only peaks residing within larger blocks of H3K27me3."

3. Page 11 "For example, DPPA5 is highly transcribed in naive PSCs and the promoter interacts with three SEs that are marked by high levels of H3K27ac and H3K4me1 (Figure 4G)." I'm not sure I understand the RNA-seq data in Fig 4G. Is DPPA5 downregulated in naive cells?

4. "Collectively, our findings have revealed that an extensive rewiring of SE – promoter interactions occurs between naive and primed PSCs," Wasn't this expected based on existing published data in this system? Maybe reformulated to articulate better the novelty?

5. "Despite the strong associations between OSN binding and active enhancers in both cell types, we found

14 that only a surprisingly small proportion (<20%) of active enhancers are shared between naive and primed PSCs

15 (Supplementary Table 3)." Is this associated with gene expression?

Reviewer #2 (Remarks to the Author):

The work by Chovanec, Collier and colleagues introduces Canvas, a computational framework to analyze genome interaction data, in particular developed and applied for promoter-capture-Hi-C. The work, which tackles some unaddressed technicalities in the field of interaction-data analysis, was applied to a comprehensive dataset of differential interactions between naïve and prime PSC. Overall, the work seems very well conducted and is well written. The new method adds to the list of existing computational tools to analyze interaction datasets. Although, one arguably could indicate that this is not totally novel (for example, the Assortativity Measure by Pancaldi and colleagues, which is referred at the end of the manuscript is conceptually similar). As the authors state, the method and the results "provide a valuable resource to investigate ... regulatory regions in the genome". I agree with such statement. I would suggest, however, the authors address some issues that, in my opinion, disfavor the overall work. Those are listed next in order of relevance.

1. Replicate-based reproducibility of the results. Although the authors show that the replicates of pHi-C have high correlation, it would be important to assess whether the networks change between replicates more than between states. This is, in my opinion, necessary. Interaction selection is not an easy task and all peak/loop detection methods suffer from reproducibility.

2. Why the authors use two different ways of normalizing the data (HOMER and Juicer? Please, clearly justify.

3. Why the authors decide to add interactions with CHICAGO score between 3 and 5 for Figures 4 and 6. This needs a strong justification.

4. Do the authors use any weight for the edges in the network layout? If not, why not? How that would change the maps?

5. The work by Pancaldi and colleagues (ref 77 in the manuscript) is only mentioned at the very end

of the manuscript (discussion). Being this a method very relevant to Canvas, it deserves that it is placed in context of the overall work.

6. Similarly, the authors could put the overall work in perspective of Mas et al 2018 <https://doi.org/10.1038/s41588-018-0218-5>. Of particular interest would be to relate their work on the observed the NOS widening of peaks for bivalent promoters.

7. Please, quantify the many statements in the text where qualitative sentences such as “slightly higher”, “tended to be larger”, or “strongly enriched” (among many others) are stated instead of numbers that the authors actually have.

8. In page 15 the authors talk about “transcriptional hubs” and in the next sentence imply that they fund those pre-formed. There is no evidence of such identified hubs being “transcriptional hubs”.

9. Is Figure 2D really necessary? (a plot for 2 numbers). In the same figure, I would suggest better explain Figure 2A in both the text and the legend.

Reviewer #3 (Remarks to the Author):

In this work, Chovanec et al. generated useful 3D genome structure data by performing Hi-C and Promoter Capture Hi-C in native PSC and primed PSC. They also described chromatin profiles and gene regulatory interactions in these two cell types. To study the role of OST in these two cell types, the author also performed the NANOG and SOX2 ChIP-seq. Overall, I found this an interesting study, and I particularly like the especially the network graph based the promoter capture Hi-c data.

Some comments for the authors to further improve the quality of this work:

- 1) Page 7, lines 1-2. The authors mentioned the insulation score of TAD boundaries was higher in primed PSCs than the naïve PSCs. Can this observation also be explained by the Canvas results?
- 2) The authors suggested several times the network scale visualization can be used to explore the 3D chromosome organization, which is very similar to Hi-C. All the network is based on the promoter capture Hi-C, I think if the author can provide a pile-up related figure to show the interaction profile of promoter capture Hi-C and Hi-C in the interaction cluster will strengthen the conclusion.
- 3) Figure 1d is complicated and not clear. The authors need to add more details.
- 4) Page 8, lines 7-8. In Fig. legend for 2D mentioned it is the total number of loops in chromosome 5, but here the author said it is long-range chromatin interaction. Need to clarify.
- 5) Page 8, lines 12-15. It is unclear what's the region authors used to show the APA plot. Based on the description from the main text, I thought all the long-range interactions were used to plot the APA. But in the figure legend, the author mentioned it is for primed-specific chromatin interaction.
- 6) Page 10, lines 11-16. The author mentioned that there are ~600-700 of SEs are shared between both cell types, but only 15% of genes were contracted by SEs in both native and primed PSCs, which is based on the promoter capture Hi-C results. it is very interesting that they share ~70% of SEs, but most of them are not regulate the same gene. Need better explanation here.
- 7) Page 11, lines 11-14. Based on the results shown in Figure 4G. It is still a significant change between native and primed PSCs in the shared enhancer targeted gene. Can these changes be explained by the interaction frequency of h3k27ac intensity?
- 8) Page 11, lines 11-14. Is there any h3k27me3 change in the example region? Because the author mentioned the H3K27me3 leads the differential long-range interaction between native and primed PSCs.
- 9) Page 13, lines 18-20. 1. In Supplementary Figure 8, there are strong and broad ATAC-seq and H3K27ac signals in the top naïve-specific OSN binding sites, but less OSN single in the corresponding region. This looks weird. 2. At the same time, if the upper panel and lower panel in Supplement Figure 8 are used the same order of naïve-specific OSN binding sites, that suggests these regions with strong active promoter mark single, but with lower OSN signal. Does this mean OSN binding is more enriched in the distal /no-H3K4me3 region?

We are delighted and very grateful to have received such considered and positive comments from all three reviewers. We respond below to each point raised. Please note that the additions made to the revised manuscript are highlighted in blue text in the accompanying manuscript file.

Reviewer #1

In this manuscript “Widespread reorganisation of pluripotent factor binding and gene regulatory interactions between human pluripotent states” by Chovanec et al. the authors report on a new way to visualize and explore functional chromatin interactions between two distinct pluripotency stages in human cells, naive and primed. This is a very interesting manuscript, of interest beyond the stem cell field. This developmental stage is unique in exploring how cellular identities are established. What I found particularly exciting is that the data suggests that the two pluripotent states have large clusters of interacting genes, which might have functional relevance, and also because forward differentiation from naive to primed is forming de novo long range interactions, specifically in bivalent regions.

I have a couple of significant issues.

1. I haven't seen any molecular experimental validation of the findings. Not being an expert in Hi-C methodologies, is there so much confidence in these pipelines that it doesn't warrant independent experimental validation of strong interactions that change between naive and primed? The data is very interesting and the visualisation is remarkable, but it would be highly desirable to have some molecular data to corroborate the large scale dataset findings, so far this is at best a potentially excellent hypothesis generation technology. For example, can one not visualize the H1 clusters, claimed to be specific for naive cells and protocadherin, claimed to be specific for primed cells, using DNA FISH? A further suggestion, if feasible: it would be interesting to see the temporal resolution of these interactions and the kinetics of this process (ie. to establish whether this is a quick change or a gradual change).

We agree with the Reviewer that additional validation experiments would help to strengthen the Hi-C-based results. As suggested, we have used three-dimensional DNA-FISH to examine an exemplar region that differs in chromatin interactions between naive and primed PSCs. Importantly, the new results presented in Supplementary Fig. 7d validate our promoter-capture Hi-C (PChi-C) data. Specifically, we show using DNA-FISH and PChi-C that in primed PSCs, *HOXD10/11* and *DLX1/2* loci are closer together compared to *HOXD10/11* and a control locus that is equidistant in the opposite direction along the chromosome. In contrast, in naive PSCs, there was no difference in the proximity between *HOXD10/11* – *DLX1/2* and *HOXD10/11* – control locus. These results confirm that long-range interactions associated with developmental genes differ between naive and primed PSCs.

2. Regarding DNA methylation and the title of the subsection “Naive-specific active enhancers are decommissioned predominantly 1 by DNA methylation” The data shows a correlation indeed, but the title would suggest that a DNMT3a/b KO or KD cell line was used to demonstrate that active enhancers are decommissioned by DNA methylation. This is a very interesting hypothesis, which would imply that in the absence of DNA methyltransferases (KO, KD or chemical inhibition) the majority of enhancers would not be decommissioned. If this is not feasible then only correlation can be claimed, which is also a significant step. The wording will then need to be adjusted.

We concur with the Reviewer. In response, we have reworded the section heading to “Decommissioning of naive-specific active enhancers correlates with the acquisition of DNA

methylation” and have toned down several of the sentences to more accurately convey the main findings.

Further questions:

3a. Regarding PChi-C, are two biological replicates sufficient? There is a very significant correlation in Fig S1A, right panel, between the two pluripotent stages. What is the statistical uncertainty that any given point is outside the difference in replicates? One possible way to express this would be, for figure 1A(left) take the difference between the read counts of the replicates on the y axis, and the average of the read counts on the x axis. The standard error is then obvious from the y axis, and any variation in standard error versus average read count can also be calculated. Do the same for 1A(middle). Now for 1A(right) you can estimate, for every point, the z-score of the difference from primed to naive. Other methods may be even more appropriate, but the essential point is that you should be able to provide a much more robust argument for the difference being claimed. If a method is embedded in the pipelines, that needs to be clearly mentioned in the main text.

The Reviewer raises an important point and we have expanded the description of how the PChi-C data were analysed in the Methods section.

We have used the CHiCAGO pipeline to call interactions in naïve or primed hPSCs, which is a state-of-the art bioinformatic method that is specifically designed to robustly identify interactions in Capture-Hi-C datasets (Cairns et al., 2016). Replicates are combined in a manner similar to the sample normalisation strategy implemented in DESeq, using sample size factors as the mean ratio of the geometric mean of each fragment pair’s count in the replicate to the geometric mean of each fragment pair’s count across all replicates. The merged counts are derived as the rounded weighted sum of counts across replicates, where the weights are the sample size factors. Interactions are called from the merged data. CHiCAGO uses a convolution background model for the background level of interactions for pooled baited and other regions, and a weighted distance-dependent multiple testing correction. The weight profile for the multiple testing correction is generated by fitting a bounded logistic curve to the observed reproducibility levels between the unmerged replicates at different distances. Based on this, a p-value is assigned to each fragment read pair (please see our response to comment #4 for more details about the p-values and interaction calling). The background model is a convolution of negative binomial (a genomic distance dependent term modelling Brownian collisions) and Poisson distributions (a distance-independent term modelling technical noise). Importantly, the weighted, distance-dependent multiple testing correction based on the reproducibility of interactions between individual replicates makes CHiCAGO robust in the presence of undersampling, in regions with low counts that are common in Capture Hi-C data, providing a rigorous statistical framework for identifying interactions.

3b. Also notice that H3K4me1_naive ChIP-seq is done in only one replicate. Can the authors comment on how they arrived to using one or two replicates per sample?

We agree with the reviewer that providing only one replicate of H3K4me1_naive ChIP-seq data in our original submission could be perceived as substandard compared to the replicated and high quality data sets that we generated in the rest of our study. We have therefore repeated the H3K4me1 experiments in the same cell lines so that we now provide two biological replicates for naive PSCs and two for primed PSCs. We used CUT&RUN instead of ChIP-Seq due to the benefits of this newer method in generating strong signals and low background. Importantly, there

was an excellent correlation between our new and old data sets (Reviewer Figure 1). Figures 3, 4, 5 and 6 and Supplementary Figures 6, 7, 8 and 9 have been updated with the new data, resulting in very minor changes and, as expected, all of the main results and findings remain the same. Providing these additional data means that all newly generated data sets in this study have been performed in two independent, biological replicates, which is standard for these assays and suitable for robust analyses.

Reviewer Figure 1. Left: H3K4me1 ChIP-seq vs CUT&RUN signals for an 80 kb region on chromosome 10. Right: correlation matrix for naive and primed ChIPseq and CUT&RUN samples (5 kb windows, log₂ RPM > 1 in any sample).

4. Related to this, in Canvas, in the network graph, what accounts for a significant interaction “each edge represents a significant interaction between nodes” in Line 23? Also, I guess I don't get a sense of the population heterogeneity. Is significance a measure of increased interaction within a population? Can heterogeneity be inferred from this data in a specific locus? Stem cells are known to be heterogeneous, particularly primed, how can one distinguish heterogeneity from a non-significant interaction?

Significant interactions shown in the network graph were identified using CHiCAGO analysis and we have added text to the results to clarify this. Briefly, CHiCAGO interaction scores correspond to $-\log$ -transformed, weighted p-values for each fragment read pair. In the weighting procedure, the weights are trained on the reproducibility of interaction calls with distance (Cairns et al., 2016). Soft-thresholding was also implemented to shift the $-\log$ -transformed, weighted p-values such that the score of zero corresponds to the probability of a true interaction in the very short range given zero reads. A CHiCAGO interaction score of 5 or above was considered significant based on previous empirical observations using other types of genomic data, such as histone marks, to select a suitable threshold (Cairns et al., 2016; Freire-Pritchett et al., 2017); thus, we use previously established CHiCAGO thresholds to call significant promoter interactions.

The reviewer raises several interesting points about the potential impact of population heterogeneity on the interaction analysis, and we now discuss this topic in the revised manuscript. If the presence of an interaction does vary depending on cell state, cell cycle or other parameters, then this signal would be averaged in the analysis. The Reviewer is therefore correct that an interaction found only in a subset of cells might be deemed as not significant based on a bulk

population analysis. In essence, CHiCAGO analysis of the PCHiC data can be viewed as a ranking measure for interaction calls and the position of an interaction within the rank will be partly dependent on the proportion of cells in the sample harbouring that interaction. Exploring the cell-to-cell variability of individual promoter interactions is an exciting topic and one that will require the development of new experimental and computational methods. We agree with the Reviewer that some conditions that are used to culture primed PSCs do lead to cell heterogeneity, although, in our experience, such heterogeneity is quite well dampened in cells that are cultured in E8 conditions, as we used in this current study.

5. NK2 cells are known to acquire tetraploidy over time (a significant fraction of the population), which I assume will impact on chromatin interactions. Have the authors mitigate against this potential problem, can they comment?

We agree that this could be a potential problem, although when we have previously examined our own naive NK2 cultures we found there was only a small percentage of tetraploid cells. However, to more stringently examine the genome stability of the cells that we used in this current project, we used the Hi-C sequencing data generated in this study to identify copy number variations. The new data, presented in Supplementary Fig. 11 and discussed in the Methods, shows that the naive NK2 cells have a normal copy number across nearly all chromosomes, and no tetraploid signatures were detected. The main alteration is on chromosome 19, and this alternation is present in both naive and primed cells. These data indicate that the PSCs used in this study have good genome stability and that tetraploidy was not present at a detectable level in the samples.

Minor points:

1. Suppl Fig 2B Differentially expressed is clear but not clear in which direction.

We have made Supplementary Fig. 2b clearer by indicating the direction of the changes.

2. "This implies that the H3K4me1/3-marked sites are protected from H3K27me3 in naive PSCs, but that H3K27me3 spreads throughout the region in primed PSCs. Genes within these regions were associated with developmental processes, with examples including DLX, GATA and HOX factors." Can one see profiles of these genes/regions, in primed and naive, to see how "commonly with H3K4me1/3-only peaks residing within larger blocks of H3K27me3."

We agree that this would be helpful. In response, we have added new genome profiles to exemplify this observation (Supplementary Fig. 6b).

3. Page 11 "For example, DPPA5 is highly transcribed in naive PSCs and the promoter interacts with three SEs that are marked by high levels of H3K27ac and H3K4me1 (Figure 4G)." I'm not sure I understand the RNA-seq data in Fig 4G. Is DPPA5 downregulated in naive cells?

We apologise this was unclear in the previous version. The RNA-seq tracks show strand-specific RNA-seq data – we now mention this specifically in the figure legend. In Fig. 4g, the RNA-seq signal below the centre line in the naive PSC panel shows that *DPPA5* is strongly expressed from the negative strand.

4. *“Collectively, our findings have revealed that an extensive rewiring of SE – promoter interactions occurs between naive and primed PSCs,” Wasn’t this expected based on existing published data in this system? Maybe reformulated to articulate better the novelty?*

We agree with the Reviewer and in response we have reworded this sentence in the manuscript. The new text reads: *“Collectively, our global and high-resolution mapping of promoter cis-regulatory interactions revealed that most SE-target genes were cell type-specific, identifying close to 1000 genes with SE interactions that differed between naïve and primed hPSCs.”*

5. *“Despite the strong associations between OSN binding and active enhancers in both cell types, we found that only a surprisingly small proportion (<20%) of active enhancers are shared between naive and primed PSCs (Supplementary Table 3).” Is this associated with gene expression?*

Yes, genes that interact with shared enhancers tend to be expressed at similar levels in naive and primed PSCs and at higher levels compared to genes that do not interact with an enhancer. These results are shown in Fig. 4c, f.

Reviewer #2

The work by Chovanec, Collier and colleagues introduces Canvas, a computational framework to analyze genome interaction data, in particular developed and applied for promoter-capture-Hi-C. The work, which tackles some unaddressed technicalities in the field of interaction-data analysis, was applied to a comprehensive dataset of differential interactions between naïve and prime PSC. Overall, the work seems very well conducted and is well written. The new method adds to the list of existing computational tools to analyze interaction datasets. Although, one arguably could indicate that this is not totally novel (for example, the Assortativity Measure by Pancaldi and colleagues, which is referred at the end of the manuscript is conceptually similar). As the authors state, the method and the results “provide a valuable resource to investigate .../... regulatory regions in the genome”. I agree with such statement. I would suggest, however, the authors address some issues that, in my opinion, disfavor the overall work. Those are listed next in order of relevance.

1. Replicate-based reproducibility of the results. Although the authors show that the replicates of pcHi-C have high correlation, it would be important to assess whether the networks change between replicates more than between states. This is, in my opinion, necessary. Interaction selection is not an easy task and all peak/loop detection methods suffer from reproducibility.

We agree with the Reviewer that this is an important point. In response, we have added several new figures to compare the networks that are generated separately using PCHi-C data from each individual replicate. Importantly, the key landmarks of the networks are very consistent between the replicates, including the entire network (Supplementary Fig. 2c), Polycomb-associated interactions (Supplementary Fig. 2d) and the changes in the node and edge characteristics for each sub-network (Supplementary Fig. 4a). Overall, these results demonstrate there is good reproducibility between replicate samples.

2. Why the authors use two different ways of normalizing the data (HOMER and Juicer? Please, clearly justify.

We apologise to the Reviewers - we made a mistake in the Methods text of the previous version of the manuscript and we have now corrected this. To clarify, HOMER normalisation was not used; instead, raw counts were used to call TADs in HOMER v4.7.

The Reviewer raises a very interesting point though and so, for completeness, we address the general topic of using different normalisation strategies in Hi-C. Both HOMER and Juicer tools have internal implementations for normalisation. Although the implementations are different, they are both matrix balancing algorithms (a matrix operation which turns all row- and column-sums of a matrix equal). We have compared the corrected matrices between HOMER and Juicer tools for our data, and this analysis shows that the corrected matrices are comparable (Reviewer Figure 2 on the following page).

Reviewer Figure 2: Scatterplot comparing the log₂ corrected counts in the coverage-corrected Hi-C matrices for chromosome 5, using JuicerTools' Knight-Ruiz matrix balancing algorithm (y-axis) against HOMER's matrix balancing (x-axis). Each dot represents a matrix element, where Hi-C data were binned at 1Mb resolution.

3. *Why the authors decide to add interactions with CHiCAGO score between 3 and 5 for Figures 4 and 6. This needs a strong justification.*

Figures 4 and 6 focus predominantly on investigating promoter–enhancer communication in naive and primed PSCs, with a view to identifying shared and state-specific gene regulatory interactions. An initial concern that we had when working on these comparisons was that our standard analysis would be too sensitive in calling differences in interactions between cell types if we applied a strict cutoff of a CHiCAGO score of >5. We therefore adopted a more cautious approach of allowing an interaction to be called if it had a CHiCAGO score of >3 in one cell type *and* a score >5 in the other cell type. That way we could be more confident that the identified differences in interactions between cell types were likely to be robust and not due to a minor sub-threshold score, such as 4.9 vs 5.0, which would be classed as a cell-type-specific without our amendments. This approach was based partly on our experience in an earlier PCHi-C project where a similar relaxed threshold was very effective (Freire-Pritchett et al., 2017). We have expanded our rationale for this analysis in the Methods.

4. *Do the authors use any weight for the edges in the network layout? If not, why not? How that would change the maps?*

We thank the Reviewer for encouraging us to look further at this topic. We did not use edge weights in the original manuscript, but this is an interesting idea and one that we have also been thinking about. In the revised manuscript, we now discuss several possibilities for how the network graphs could be customised to include edge weight information and we present new data to exemplify this (Supplementary Fig. 10). Overall, including edge weights does not have much of an impact on the network layout, but we hope that by providing these new data the readers can see how Canvas can be adapted and extended.

5. The work by Pancaldi and colleagues (ref 77 in the manuscript) is only mentioned at the very end of the manuscript (discussion). Being this a method very relevant to Canvas, it deserves that it is placed in context of the overall work.

We agree with the Reviewer and we have added a new paragraph to the introduction where we highlight the Pancaldi et al. publication and other studies in this area.

6. Similarly, the authors could put the overall work in perspective of Mas et al 2018 <https://doi.org/10.1038/s41588-018-0218-5>. Of particular interest would be to relate their work on the observed the NOS widening of peaks for bivalent promoters.

The Mas et al. 2018 paper describes how MLL2 regulates the chromatin accessibility and long-range interactivity of bivalent promoters in mouse PSCs. Related work by Bing Ren and colleagues showed that MLL3/4 orchestrate long-range interactions at enhancers (Yan et al., 2018). We now discuss these studies in the context of our own work.

7. Please, quantify the many statements in the text where qualitative sentences such as “slightly higher”, “tended to be larger”, or “strongly enriched” (among many others) are stated instead of numbers that the authors actually have.

We have been through the manuscript text and have added quantitative descriptions or re-written sentences where appropriate. Examples include:

“We additionally found, however, that the insulation score of TAD boundaries was higher in primed compared to naive PSCs (by 15%; naive: 0.80, primed: 0.92; Supplementary Fig. 3b), suggesting there are differences in TAD boundary strength between the two cell types.”

“Only one-third of these regions were classified as bivalent in naive compared to primed PSCs (208/1000, naive; 655/1000, primed) and very few of these interactions were detected in naive PSCs (n=37/1000; Fig. 3a), suggesting that they are formed de novo upon the transition from naive to primed pluripotency.”

“Interestingly, essentially all OSN sites in both cell types contained enhancer chromatin signatures including H3K27ac (92 % with log₂ RPM > 0) and H3K4me1 signals (98 %) and open ATAC-seq regions (84 %, Supplementary Fig. 9a). A small subset of OSN regions was also positive for H3K4me3 (12 % with log₂ RPM > 0) while showing lower H3K4me1 (Supplementary Fig. 9a), suggesting that they have promoter activity.”

“Furthermore, when OSN occupancy was lost during the transition from a naive to a primed state, 74% (n=4294) of all interactions at these sites were also lost, and when OSN was gained, 86% (n=247) of all interactions at these regions were gained (Fig. 6d; p<0.0001; Chi-squared test).”

“By re-analysing ChIP-seq data, we found that the naive-associated transcription factor TFAP2C was enriched (> 3-fold) at OSN-bound regions in naive PSCs (Fig. 6g) and indeed ~20% of these regions contained TFAP2C peaks.”

8. In page 15 the authors talk about “transcriptional hubs” and in the next sentence imply that they fund those pre-formed. There is no evidence of such identified hubs being “transcriptional hubs”.

We have modified the sentence to remove the term “transcriptional hubs”.

9. Is Figure 2D really necessary? (a plot for 2 numbers). In the same figure, I would suggest better explain Figure 2A in both the text and the legend.

We would prefer to keep Fig. 2d because it does help to visually show the key difference in chromatin loops between naive and primed hPSCs, particularly for those readers who tend to focus mainly on the figures without referring to the main text.

As suggested, we have modified the main text, the figure legend and the figure to better explain Fig. 2a.

Reviewer #3

In this work, Chovanec et al. generated useful 3D genome structure data by performing Hi-C and Promoter Capture Hi-C in native PSC and primed PSC. They also described chromatin profiles and gene regulatory interactions in these two cell types. To study the role of OST in these two cell types, the author also performed the NANOG and SOX2 ChIP-seq. Overall, I found this an interesting study, and I particularly like the especially the network graph based the promoter capture Hi-c data.

Some comments for the authors to further improve the quality of this work:

1) Page 7, lines 1-2. The authors mentioned the insulation score of TAD boundaries was higher in primed PSCs than the naïve PSCs. Can this observation also be explained by the Canvas results?

In the original manuscript, we discovered that sub-networks identified in the promoter-capture Hi-C data correspond closely to TADs that were defined using the Hi-C results. As the Reviewer mentions, we also found that the insulation score of TAD boundaries was higher in primed compared to naive PSCs. Calculating insulation scores of the sub-networks is challenging because most of the sub-networks are not connected to each other because the whole network is formed from numerous discrete and disconnected sub-networks rather than one continuous set of regions. Unfortunately, there are too few sub-networks that retain information about neighbouring boundaries for us to measure insulation scores in this way.

2) The authors suggested several times the network scale visualization can be used to explore the 3D chromosome organization, which is very similar to Hi-C. All the network is based on the promoter capture Hi-C, I think if the author can provide a pile-up related figure to show the interaction profile of promoter capture Hi-C and Hi-C in the interaction cluster will strengthen the conclusion.

We agree with the Reviewer that this analysis will further strengthen our conclusions and so we now provide pile-up figures to show the interaction profiles of PCHi-C and Hi-C. We have included plots at 25 kb resolution based on two different viewpoints: all significant interactions and all significant interactions within one interaction cluster (Supplementary Fig. 1c). The results show that interaction signals are present in both the PCHi-C and Hi-C datasets, and, as expected, the signals are stronger in the PCHi-C data due to the higher coverage at these interacting regions compared with the respective Hi-C samples. In the revised manuscript, we have also included more information and relevant citations to point the reader towards previous studies that have comprehensively compared Hi-C and PCHi-C data sets.

3) Figure 1d is complicated and not clear. The authors need to add more details.

As requested, we have revised Fig. 1d and expanded the figure legend to add more details.

4) Page 8, lines 7-8. In Fig. legend for 2D mentioned it is the total number of loops in chromosome 5, but here the author said it is long-range chromatin interaction. Need to clarify.

Good point - thank you. We have changed the wording in the figure legend so that the terms are now consistent with the main text.

5) Page 8, lines 12-15. It is unclear what's the region authors used to show the APA plot. Based on the description from the main text, I thought all the long-range interactions were used to plot the APA. But in the figure legend, the author mentioned it is for primed-specific chromatin interaction.

We apologise that this was not clear in the manuscript. We have now revised the figure and text to show all long-range interactions (Fig. 2f).

6) Page 10, lines 11-16. The author mentioned that there are ~600-700 of SEs are shared between both cell types, but only 15% of genes were contracted by SEs in both native and primed PSCs, which is based on the promoter capture Hi-C results. It is very interesting that they share ~70% of SEs, but most of them are not regulate the same gene. Need better explanation here.

We agree with the Reviewer that this finding is very interesting and we have now examined the topic further and provided additional interpretation. In our new analysis, we show that there are a large number of gene promoters (n=633) that interact with a shared SE only in naive PSCs (Supplementary Fig. 8a). In contrast, relatively few promoters (n=250) interact with a shared SE only in primed PSCs (Supplementary Fig. 8a). Based on this, we hypothesised that SEs might interact with more promoters in naive PSCs compared to primed PSCs, however, we found that the number of interacting promoters per SE was very similar between the two cell types (Supplementary Fig. 8b,c). These results suggest that the presence of a SE does not necessarily lead to an interaction between a SE and its target gene promoter, and that currently unknown cell type-specific regulators are additionally required to promote SE – promoter communication. Notably, these observations are consistent with our prior study that revealed dynamic promoter – SE interactions between mouse pluripotent states (Novo et al., 2018). Together, these results further underscore the importance of integrating chromatin state annotations with interaction data to develop a more complete understanding of gene regulation, and we anticipate that *Canvas* will be a valuable approach to facilitate this.

7) Page 11, lines 11-14. Based on the results shown in Figure 4G. It is still a significant change between native and primed PSCs in the shared enhancer targeted gene. Can these changes be explained by the interaction frequency of h3k27ac intensity?

Yes, we agree with the Reviewer that it is very plausible that some of the interactions shown in Fig. 4g could be explained by differential H3K27ac. We have added text to the manuscript to highlight the changes in H3K27ac levels. It is also important to note that the capture-based Hi-C approach used in our study ensures that the detection of interactions is independent of changing histone modifications or chromatin-binding factors, in contrast to other assays, such as ChIA-PET and HiChIP.

8) Page 11, lines 11-14. Is there any h3k27me3 change in the example region? Because the author mentioned the H3K27me3 leads the differential long-range interaction between native and primed PSCs.

We have updated Fig. 4g to include H3K27me3 tracks. These data show that there is very little H3K27me3 in either naive or primed PSCs in this region, suggesting that the differential long-

range interactions at the *DPPA5* promoter are mediated by alternative processes, such as changes in enhancer activity. Interestingly, when examining H3K27me3 at an additional exemplar region (*TBX3*; Supplementary Fig. 8e), we found that this locus switches from predominantly active chromatin marks in naive PSCs to high levels of H3K27me3 in primed PSCs. In primed PSCs, the *TBX3* promoter forms long-range interactions with several other H3K27me3-marked sites including *TBX5* and *LHX5*, whereas in naive PSCs the *TBX3* promoter interacts mainly with putative enhancers. Thus, the reviewer correctly anticipates changes in H3K27me3 that could lead to differential long-range interactions between naive and primed PSCs.

9) Page 13, lines 18-20.

1. In Supplementary Figure 8, there are strong and broad ATAC-seq and H3K27ac signals in the top naive-specific OSN binding sites, but less OSN signal in the corresponding region. This looks weird.

We believe that the regions identified by the Reviewer correspond to promoter regions, and we discuss this comment below.

2. At the same time, if the upper panel and lower panel in Supplement Figure 8 are used the same order of naive-specific OSN binding sites, that suggests these regions with strong active promoter mark signal, but with lower OSN signal. Does this mean OSN binding is more enriched in the distal /no-H3K4me3 region?

We thank the Reviewer for encouraging us to look further at the distribution of OCT4, SOX2 and NANOG (OSN). Indeed, we find that OSN signal in naive and primed PSCs is higher at distal sites compared to promoter regions, and we show these new data in Supplementary Fig. 9b. This finding is consistent with the results in Fig. 6a, which shows the occupancy of OSN at different genomic regions with a strong association to active enhancers. We have also taken the opportunity to clarify in the figure legend that the two rows in Supplementary Fig. 9a (formerly Supplementary Fig. 8) form one figure and should be viewed together.

REVIEWERS' COMMENTS

Reviewer #1 (Remarks to the Author):

The authors have addressed my critical points: they have validated their PCHi-C data using DNA FISH, which looks convincing and interesting; have rephrased the interpretation around the role of DNA methylation and have increased the number of replicates where needed with CUT&RUN. One note around tetraploidy, in my understanding one cannot infer levels of tetraploidy from CNV data, when it's simply a genome duplication. Is that right? The authors have previously assessed their own naive cultures (with FLOW?), also, they present DNA FISH data in naive cells so they will be able to derive a percentage. (In fig S11, how was the ratio calculated, ratio to what reference genome or sample?)

Reviewer #2 (Remarks to the Author):

I have no additional criticisms and thank the authors for addressing all my original ones.

Reviewer #3 (Remarks to the Author):

The authors have addressed all my questions. This is an excellent study and I recommend it for publication in Nature Communications.

Signed by: Feng Yue

Reviewer #1:

The authors have addressed my critical points: they have validated their PCHI-C data using DNA FISH, which looks convincing and interesting; have rephrased the interpretation around the role of DNA methylation and have increased the number of replicates where needed with CUT&RUN.

One note around tetraploidy, in my understanding one cannot infer levels of tetraploidy from CNV data, when it's simply a genome duplication. Is that right? The authors have previously assessed their own naive cultures (with FLOW?), also, they present DNA FISH data in naive cells so they will be able to derive a percentage. (In fig S11, how was the ratio calculated, ratio to what reference genome or sample?)

AU: As the reviewer suggests, we now include the percentage of tetraploid cells that were detected using DNA FISH data. We have added the following sentence to the methods: "We also examined the chromosome copy number of cells by DNA FISH, which showed <10% of naive PSCs were tetraploid." In Fig. S11, the ratio is sample to reference genome.

Reviewer #2:

I have no additional criticisms and thank the authors for addressing all my original ones.

Reviewer #3:

The authors have addressed all my questions. This is an excellent study and I recommend it for publication in Nature Communications.

Signed by: Feng Yue